# Large-scale englacial folding and deep-ice stratigraphy within the West Antarctic Ice Sheet

Neil Ross[1], Hugh Corr[2], and Martin Siegert[3]

[1]School of Geography, Politics and Sociology, Newcastle University, UK
[2]British Antarctic Survey, High Cross, Cambridge, UK
[3]Grantham Institute and Department of Earth Science and Engineering, Imperial College London, UK

**Correspondence:** Neil Ross (neil.ross@newcastle.ac.uk)

**Abstract.** It has been hypothesized that complex englacial structures identified within the East Antarctic and Greenland ice sheets are generated by: (i) water freezing to the ice-sheet base, evolving under ice flow; (ii) deformation of ice of varying rheology; or (iii) entrainment of basal material. Using ice-penetrating radar, we identify a widespread complex of deep-ice facies in West Antarctica that exist in the absence of basal water. These deep-ice units are extensive, thick (>500 m), and incorporate multiple highly reflective englacial layers. At the lateral margin of an enhanced flow tributary of the Institute Ice Stream, these units are heavily deformed and folded by the action of lateral flow convergence. Radar reflectivity analysis demonstrates that the uppermost reflector of the deep-ice package is highly anisotropic, representative of ice with a strong preferred crystal fabric and, consequently, will have a different rheology to the ice above and below it. Deformation and folding of the deep-ice package is an englacial response to the combination of laterally-convergent ice flow and the physical properties of the ice column.

## 1 Introduction

Recent advances in radar technology have enhanced the quality and resolution of ice-penetrating radar data, revealing complex structures in the lower ice sheet column. Thick deep-ice units have been imaged above the Gamburtsev Mountains in East Antarctica (Bell et al., 2011; Wrona et al. 2018) whilst in Greenland units heavily deformed by ice flow have been discovered (NEEM community members, 2013; Bell et al., 2014; Bons et al., 2016). These have been interpreted as the product of the accretion of basal water (Bell et al., 2011), deformation caused by varying rheology of the ice column (NEEM community members, 2013; Bons et al., 2016), or a combination of both (Bell et al., 2014; Wrona et al., 2018). Only a few studies (e.g. Wolovick et al., 2014, 2016; Leysinger Vieli et al., 2018) have assessed the impact of such units on ice-sheet flow and dynamics, and there are only limited reports of the existence of thick deformed deep-ice units in West Antarctica (Jacobel et al., 1993; Siegert et al, 2004). Englacial seismic boundaries in deep-ice across the West Antarctic Ice Sheet have been reported (Bentley, 1971; Horgan et al., 2011). Once interpreted as sediments within the ice (Bentley, 1971), it is now understood that these reflections represent a shift in the orientation of the ice crystal fabric (Gow and Williamson, 1976; Horgan et al., 2011; Robin and Millar, 1982). The spatial extent and significance of such fabrics in West Antarctica, and of deep ice units in general, have not been determined, however.

Here, we present ice-penetrating radar data revealing an extensive package of deep-ice facies locally characterized by a zone of large-scale folding at the lateral margin of a tributary of convergent enhanced (i.e. >25 ma$^{-1}$) ice flow (Joughin et al., 1999; Bamber et al., 2000). Like a structural geology problem (MacGregor et al., 2015; Hudleston, 2015), such folds can only be explained by the deformation of ice with contrasting physical properties near the base of the ice sheet. Evidence of variability in physical properties is consistent with ice-penetrating radar observations of a widespread englacial layer characterized by a strongly anisotropic ice crystal fabric, as postulated for ice folds in Greenland (Bons et al., 2016).

## 2 Methods

Airborne 150 MHz ice-penetrating radar data were acquired over the Institute and Möller ice streams (IIS, MIS) of West Antarctica in 2010/11 (Ross et al., 2012; Jeofry et al., 2018a; Ashmore et al., 2020). Coverage extended from the ice stream grounding zone to the ice divide (Figure 1a). A high-resolution grid with an across ice flow line spacing of 7.5 km, and tie lines at 25 km spacing, was acquired over the central parts of the ice stream catchments.

Data acquisition, processing and analysis of the radar data is described fully by Jeofry et al. (2018a) and is briefly summarized here. Range from the aircraft to the ice surface was determined using the aircraft radar altimeter or LiDAR measurements. Ice thickness was determined by multiplying the two-way travel-time between the picked ice surface and the picked reflection from the ice sheet bed by 0.168 m ns$^{-1}$ and applying a 10 m correction for the firn layer (Jeofry et al., 2018a). Bed elevation was calculated by subtracting ice thickness from ice surface elevation. Internal ice sheet layering, and the determination of fold axis positions, were visualized and manually picked using geophysical software. 2D synthetic-aperture radar (SAR)-processed radargrams are displayed with a natural logarithm applied to enhance reflections at depth. The data used were from the chirp mode of the PASIN system (Jeofry et al., 2018a; Ashmore et al., 2020). We did not use PASIN pulse channel data in this study.

Analysis of the returned radar power of internal reflectors at flight line crossover points was implemented to evaluate possible anisotropy within the ice column. This was possible due to the stepped nature of the aerogeophysical survey (i.e. most of the survey was flown in a series of blocks of constant aircraft elevation), implemented for acquisition of high-quality gravity data (Jordan et al., 2013), that meant that aircraft-to-ice surface range was consistent between flight lines at many crossover locations. At those cross-over points where the aircraft-to-surface range was the same in perpendicular survey lines, we stacked radar traces over 41 traces (∼400 m along track) and determined the returned integrated power down ice, between samples 200 and 1400 (two-way travel time of 54,545 ns) in the across- and along-ice flow directions.

Though the survey grid design allows us to undertake a detailed analysis of the pseudo-3D englacial structure of the IIS, the grid layout and orientation of the 2D survey flights was not aligned exactly along and across ice flow. This oblique offset can have implications for the 2D imaging of 3D englacial structures (Bingham et al., 2015). In general, however, our survey grid is approximately aligned along and across flow throughout most of our area of interest. As a result, we refer from this point onwards to 'across-' and 'along-' flow flight lines, although we acknowledge that in many places the flight lines are slightly oblique to both along and across ice flow vectors.

## 3 Observations

### 3.1 Ice-sheet stratigraphy and physical properties

The radar data reveal an extensive deep-ice unit with a distinctive ice-sheet stratigraphy across an area >250,000 km$^2$ (Figure 1). The unit extends from the Ellsworth Subglacial Highlands (Ross et al., 2014; Winter et al., 2015; Goldberg et al., 2020) to a region of thin cold-based ice between the MIS and Foundation Ice Stream (Bingham et al., 2015), and from the ice-stream trunk (Siegert et al., 2016) to near the Weddell-Ross ice divide (Figure 1a-c). The stratigraphy of the ice sheet column comprises a high-reflectivity upper unit, typically 750-1500 m in thickness, with thinly-stratified minimally-disrupted layering, typical of meteoric ice, overlying a low-reflectivity lower-ice column up to ∼1000 m thick (Figure 2). The boundary between these two stratigraphic units is often sharp and associated with either a distinctive thick (up to 200 m) band of high-reflectivity (R1) or a marked reduction in returned radio-wave energy, depending on radar instrument orientation (Figures 3 and 6). Beneath R1, within the low-reflectivity lower-ice column, a second highly reflective englacial reflection (R2) is observed. R2 sometimes diverges and bifurcates, becoming a series of 3-4 layers (Figure 2b). Here, for simplicity, we refer to R2 as a single 'reflection package'. Between R1 and R2, a series of very thin and very low-reflectivity reflections can be imaged, but below R2 (i.e. between it and the bed) layering is typically absent except for occasional, localized, near-bed reflections (Figure 2c).

The deep ice unit is extensively folded (Figures 3 and 4) at the lateral boundary of an enhanced flow tributary (Figure 1c) that feeds the IIS (Figures 1b-c, 2a). The geometry and structure of the deep ice unit that includes R1 and R2 differs markedly in three dimensions in this area. Radar data orthogonal to ice flow (Figure 3a and 4) show substantially folded layering (with amplitudes of up to 40% of the ice column in places), with clear evidence of a series of bed-unconformable anticlines and synclines (Figure 3a, 3b, 4). Along ice flow, however, the same unit is characterized by more subdued reflections, with bed-conformable undulations rather than folds (Figure 2b-c, 3c and 4). The deep-ice folds we observe are comparable in scale to those within the onset zone and trunk of Petermann Glacier in Greenland, (Bons et al., 2016) rather than the higher-frequency, lower-amplitude buckling often associated with enhanced ice flow (Conway et al., 2002; Ng and Conway, 2004).

The structure of the upper part of the ice sheet is highly affected by the presence and folding of the deep ice unit below R1 and does show evidence for higher-frequency folding. The upper ice column is disrupted by 'whirlwind' or 'tornado' features (Karlsson et al., 2009; Winter et al., 2015) caused by radar energy dispersion through focused SAR processing of sloping and buckled layers in airborne radar data (Holschuh et al., 2014) (Figure 3a and 3b). Whirlwinds have been identified as a signature of either structural disruption of the ice by enhanced flow, or slow ice flow over rugged basal topography. In our data, whirlwinds tend to be vertical to sub-vertical throughout the majority of the ice column but display a prominent deviation from vertical (i.e. they bend) at depth, specifically in the low reflectivity unit between R1 and R2 (Figure 3a and 3b). The whirlwinds approach and show greatest deviation towards R2, yet never cross this reflection boundary (Figure 3a and 3b). Due to their 3D geometry, whirlwinds are visible in data acquired perpendicular and sub- perpendicular to ice flow, but not in radar data acquired parallel to it (Figure 3) (Bingham et al., 2015).

### 3.1.1    Relationship between englacial folding and ice flow

A close spatial coincidence between the deep-ice folds and the onset of enhanced ($\sim$25 ma$^{-1}$) flow in the IIS catchment exists (Figure 1d). Five anticlines and synclines with a wavelength of 3-5 km are directly associated with the lateral boundary of an enhanced flow tributary of IIS (Bamber et al., 2000). These folds have axes oriented precisely along flow and are mappable in more than twenty across-ice-flow flight lines over a distance of $\sim$150 km (Figures 1d and 4). Due to lateral ice-flow convergence, the amplitude and wavelength of the folds associated with the shear zone increase and decrease, respectively, down-ice

flow until high ice velocities (>150 m yr$^{-1}$) and associated strain rates, and bright basal reflections, make identification in the main trunk of the IIS difficult (Figure 1c and 5). The folds are highly non-bed-conformable, despite the bed topography being notably flat (Rose et al., 2015) or characterized by long wavelength topographic variation (Figures 3a and 3b).

### 3.2    Down-ice propagation and glaciological implications of the englacial folds

Though eventually the englacial radar reflection signature of the folds is significantly modified by highly convergent fast flow
in the IIS trunk, mapping of the fold axes combined with surface observations from RADARSAT and MODIS (Figure 5) demonstrates an association between the most prominent fold (Figure 3d) and a well-defined surface flow stripe (Figure 5a) (Glasser and Gudmundsson, 2012; Ely and Clark, 2016) that extends from the onset zone where ice is typically flowing at 50-125 m yr$^{-1}$, into the trunk, through the ice stream grounding zone, and onto the Ronne Ice Shelf (Figure 5b). Radar data oriented obliquely to the ice flow direction (Figure 5c and d) demonstrate that this fold, and the surface flow stripe, are associated with
a significant shift in basal reflectivity (Siegert et al., 2016), suggesting that glaciological processes in the onset zone of IIS, of which the fold is a product, are likely to play an important role, alongside in situ subglacial topography and geology, in determining the compartmentalization and nature of ice flow within the IIS trunk (cf. Siegert et al., 2016; Jeofry et al., 2018b). The IIS trunk is divided into at least two flow compartments that relate to upstream tributary flow (Figures 5a and b), with crevasse patterns on the surface of the ice stream trunk either side of the flow stripe evidencing very different strain regimes
(Figure 5a). We also observe that the surface feature above the englacial fold coincides with the grid-SW margin of the the ice plain at the IIS grounding zone (Fricker and Padman, 2006). The fold may therefore play a role in buttressing the IIS, by influencing the position and form of the ice plain.

### 3.3    Returned power of englacial layers

R1 is often characterized by pronounced anisotropic reflections throughout the survey area. To demonstrate this anisotropy, we
investigated the returned power of R1 in and around the broad enhanced flow tributary of the IIS (Figures 6b). At survey crossover points within a 127.5 km x 75 km zone, R1 often displays clear anisotropy, with much stronger (typically 10 dB or more) returned radar power in the along ice flow direction than in radar survey data across ice flow (Figures 6 and Supplementary Figures 1-4). This anisotropy is similar in magnitude to that observed in East Antarctica as ice flows over large subglacial obstacles (Fujita et al., 1999; Siegert and Fujita, 2001; Wang et al. 2018); the explanation being that enhanced stresses on the
stoss-face of basal hills lead to exaggerated strain in specific 'soft' layers (of glacial as opposed to interglacial age).

## 4 Discussion

### 4.1 Physical explanations for englacial reflections

There are several possible explanations for the reflectivity of R1 and R2 including: (i) constructive interference from a series of multiple thin layers (Harrison, 1973; Siegert et al., 1999); (ii) preferred ice-crystal orientation fabrics (corresponding to power anisotropy for a specific radar layer) (e.g. Matsuoka et al., 2004; Eisen et al., 2007); (iii) Birefringent propagation (corresponding to power oscillations as a function of ice depth) (Fujita et al., 2006); and (iv) an abrupt spike in the conductivity of the ice column associated with the deposition of volcanic ash (Paren and Robin, 1975; Corr and Vaughan, 2008). These explanations are not mutually exclusive however, and it may be that more than one may act in combination. However, because we observe that the strength of the returned energy from R1 is highly anisotropic, with higher reflectivity in the along-flow orientation (Figures 2, 3, 4, 6 and Supplementary Figures 1-4), we conclude R1 is most likely caused by (ii) or (iii) (Fujita et al., 1999; Wang et al., 2018). The depth of R1 certainly rules out ice-density fluctuations. In general, it is difficult to separate-out (ii) and (iii). However, for the case of R1, we can assume that anisotropic scattering is the dominant cause of power anisotropy (the along and across flow profiles have similar long-wavelength behavior and the power anisotropy is localized for a set of radar layers). We therefore attribute R1 to a crystal orientation fabric.

Radar reflection anisotropy associated with crystal orientation fabric has been verified by ice core evidence from Antarctica and Greenland (Eisen et al. 2007, Drews et al. 2012; Li et al. 2018; Montagnat et al., 2014). Deep-ice anisotropic scattering has been observed in convergent ice flow zones, like our study area, in East Antarctica (Matsuoka et al., 2003; Matsuoka et al., 2004). In those studies, anisotropic englacial reflections were attributed to stacked alternating layers of single pole and vertical girdle fabrics observed in the Dome F ice core. Such a model is consistent with our radar observations and ice core observations elsewhere in West Antarctica. A single maximum crystal orientation fabric distribution (i.e. with a fabric characterised by strong vertical c-axes), typical for simple shear would not result in anisotropic scattering, as layer reflectivity would be the same in different survey orientation. A vertical girdle fabric on the other hand is consistent with anisotropic layer reflectivity, as crystals would have an oriented preferred fabric that would likely induce a backscatter response. Evidence for down-ice column evolution of crystal fabric (i.e. from isotropic to anisotropic, and then back to isotropic at depth) is observed in ice cores from West Antarctica (e.g. Gow and Williamson, 1976; Gow and Meese, 2007; Fitzpatrick et al., 2014), with anisotropic crystal fabrics typically associated with ice of last glacial age. However, as stated above, an anistropic crystal fabric (i.e. with a strong single vertical maxima) would not result in an anistropic radar response, so these gradual down-core changes cannot be the explanation for R1. In the Byrd core however, there is evidence for sharply alternating crystal fabrics (i.e. narrow cone to distributed cone and back again) associated with cloudy bands (1-60 mm thick) of glacial-age ice that incorporate tephra (Gow and Williamson, 1976; Horgan et al., 2011). Abrupt alternations in crystal fabric such as these are akin to those proposed as the cause of anisotropic radar scattering in East Antarctica (Fujita et al. 2003; Matsuoka et al., 2003; Matsuoka et al., 2004). Assuming that the cloudy bands in the Byrd core represent the same stratigraphy as R1, then this is a plausible explanation for the radar reflection anisotropy of this layer. The anisotropy cannot be due to directional roughness of layer reflectivity, as the anisotropy is unique to specific layers (Figures 6a and Supplementary Figures 1-4). R2 is also a prominent and strong reflection

(Figures 2, 3, and 4), but unlike R1 it is not characterized by anisotropic reflectivity (Figure 6). We consider R2 to represent a layer with a discretely high conductivity, similar to the bulk of internal layers in Antarctica (Siegert, 1999). The anomalously high reflectivity of R2 may represent a pronounced acidity spike, or multiple spikes, in the stratigraphy.

## 4.2 Formation of the deep ice unit

The 'freeze-on' hypothesis for the formation of basal ice (Bell et al., 2011) cannot explain the deep-ice unit and incorporated
layers (i.e. R1 to the ice sheet bed) for four reasons: (1) unlike Greenland, West Antarctica lacks input of water from surface ablation processes, so the water would need to be derived entirely from subglacial melt – across the majority of the area covered by R1 and R2 the ice is typically thin (i.e. <1.5 km), slow flowing (i.e. <25 ma$^{-1}$) and likely frozen to the bed; (2) in the central IIS catchment, where the bed is wet, the ice is relatively fast-flowing so the basal ice observed there could not have formed by freeze-on as it would advect down-flow before a significant thicknesses of ice could develop (without unrealistically high rates
of freezing) (cf. Dow et al., 2018); (3) IIS has a well-defined and efficient topographically-constrained subglacial drainage network, without widespread stores of subglacial water to act as water sources (Wright and Siegert, 2012; Jeofry et al., 2018a, 2018b); and (4) although during more extensive glacial conditions hypothesized subglacial lakes in the upper catchment of IIS-MIS associated with thicker ice and a low gradient ice sheet surface slope (cf. Livingstone et al., 2013) could provide subglacial water for basal 'freeze on', we observe the deep-ice unit in locations far removed from potential subglacial meltwater pathways
(e.g. Winter et al., 2015; Bingham et al., 2015) (Figures 1 and 2). Proposed active subglacial lakes in the IIS-MIS catchment investigated with ice-penetrating radar demonstrate little evidence for significant ponding of subglacial water (Siegert et al., 2014, 2016), with no apparent connection to the englacial folds in these locations.

Given the consistent stratigraphic position of R1 and R2 in all areas of the IIS-MIS they are found (e.g. Figures 1 and 2), regardless of bed conditions and flow speeds, we conclude that the structures and lateral extent of the deep-ice structures must
be the result of the deformation and localized folding of meteoric ice. Further, given the radar anisotropy observed, the most likely explanation for the folds is that they are caused by a combination of convergent ice flow and the distinct physical (i.e. varying crystal orientation fabric), and subsequent rheological, properties of the band of ice associated with R1.

## 4.3 Physical properties and rheology of the folds: modulation of ice flow

The most prominent fold observed is consistently associated with the boundary of enhanced ice flow and the ice stream trunk
surface flow stripe (Figures 1c, 3a-b and 5).

This spatial correspondence between the fold and shear margin is remarkable and may suggest that the folding of the deep ice modulates the position of the shear margin and controls trunk flow. The fold may therefore play an important role in the ice dynamics of the IIS-MIS catchment. The simplest explanation is that the core of the fold contains higher viscosity material, be that ice and/or sediment, which is less conducive to deformation than adjacent ice. The high viscosity material resists ice flow,
leading to (relatively) fast ice flow to one side and slower flow on the other, where up to four other folds are present. Evidence supporting the core of the primary fold being more resistant to flow exists in observations of ice surface 'bumps' directly above the largest anticline in some across flow survey lines.

The core of the most prominent anticline contains a consistently bright 'hand-shaped' reflection at the bed, several hundred meters high and wide (Figures 3a, 3d, 4 and 5) that allows its unequivocal identification (Figure 5). This reflection is observed, below R2, in more than fifteen across-flow lines and in the survey lines cutting obliquely across the ice stream trunk (Figure 5). Along-flow, the feature is a set of bed-conformable reflectors (Figure 3c). Our interpretation of this reflection is that it represents sediment drawn up into fractures in the base of the ice sheet during compressional fold formation (cf. Winter et al., 2019); such an inclusion of sediment would significantly alter the rheology of the core of the anticline. This may determine the position of the lateral margin of the tributary of enhanced ice flow, the different 'compartments' of ice stream trunk flow, and the position of the IIS ice plain.

## 4.4 The influence of climate history on ice sheet stratigraphy, structure and rheology

The radar reflections that characterize the IIS-MIS deep-ice unit have distinct signatures that are comparable to deep-ice reflections recorded beneath the central Greenland Ice Sheet where, because of strongly contrasting physical and rheological properties of glacial and interglacial ice, deep ice of Eemian age and older, is folded, sheared and overturned (NEEM community members, 2013; Bell et al., 2014; Bons et al., 2016; MacGregor et al., 2015). Similarly, it is possible that the varying reflection properties of the IIS-MIS deep-ice package are associated with boundaries or interfaces between ice accumulated during contrasting climatic periods (i.e. glacial/stadial and interglacial/interstadial periods). The last deglaciation in West Antarctica was characterized by a rapid increase in the rate of surface accumulation (WAIS Divide Project Members, 2013), and a 192-year period of acid deposition associated with major volcanic activity (Gow and Williamson, 1976; McConnell et al., 2017). A new radiostratigraphy across West Antarctica (Ashmore et al., 2020) enables us to infer the likely age of the deep ice units in the IIS-MIS. The best 'direct' constraint on the age of the deep-ice package is from the dated radar englacial layer tied to the Byrd ice core (Siegert and Payne, 2004; Ashmore et al., 2020). Linking the radar stratigraphy of the IIS-MIS catchment to the radar transect of Siegert and Payne (2004) where our 2010-11 survey data intersects their profile (Ashmore et al., 2020), indicates that the transition between the upper undisrupted high reflectivity ice and the lower deep-ice unit (i.e. R1) can be dated to approximately 16 ka. There are some potential uncertainties associated with this correlation (e.g. resolution of the Byrd ice core, the resolution, geolocation and vertical and along track sampling of the SPRI-NSF-TUD RES data etc.). However, assuming this correlation to be correct, the low-reflectivity zone between R1 and R2 is of last glacial age, with the potential for even older ice in places. As glacial ice tends to be rheologically 'softer' than interglacial ice, enhanced flow often occurs in such layers (Paterson 1991; Pettit et al., 2014; NEEM community members, 2013). The strongly preferred ice crystal fabric of R1 could be the product of differential shear associated with ice sheet physical properties produced by: (i) the abrupt change in WAIS accumulation (WAIS Divide Project Members, 2013); and/or (ii) the volcanic deposit known as 'Old Faithful' (Jacobel and Welch, 2005), both associated with the termination of the last glacial period in West Antarctica. R2 would then represent ice older, potentially significantly older, than the Last Glacial Maximum. The bifurcation of R2 into multiple layers (Figure 2b) may be of note here; multiple tephras dating to the last glacial cycle (i.e. at 22.3 ka, 32.4 ka and 44.8 ka) have recently been recorded in parts of West Antarctica (Iverson et al., 2017). Older widespread West Antarctic tephras from Marine Isotopic Stages 6 and 5 are also recognised in marine sediments from the continental shelf (Hillenbrand et al., 2008).

Englacial processes similar to those responsible for the formation of the deep-ice unit in the IIS-MIS catchment are likely to occur elsewhere in Antarctica. Radar data from the Siple Coast region record the presence of a major fold and upwarping of ice, for example (Siegert et al., 2004). The formation of this fold was originally ascribed to a Holocene-age ice flow reorganization, but formation through deformation and folding of deep ice of varying rheological properties by convergent ice flow is a viable explanation prior to ice-flow direction change.

## 5   Conclusions

We have demonstrated the presence of an extensive package of deep-ice units beneath the Institute and Moller ice streams. At least one layer in the deep ice has physical properties (i.e. ice crystal orientation fabric, rheology) significantly different to the upper ice column. At the lateral boundary of the onset of enhanced ice flow of IIS, where ice flow is convergent, these deep-ice units have been heavily deformed. Deformation has led to the development of large-scale englacial folds that may modulate ice-stream position, structure and dynamics. Due to the extensive spatial extent of the deep-ice unit, and its vertical and horizontal variability in crystal fabric and rheology, such units have important implications for assumptions underlying our understanding of ice sheet tributary and trunk flow. Our results confirm that ice flow across the bulk of West Antarctica, and potentially other ice masses, is more complex than is currently incorporated within the set-up and application of many numerical ice-sheet simulations used to determine ice dynamics and predict global sea level. As future ice flow models with greater computational power and grid resolution incorporate the effects of rheology on ice dynamics, they must ensure that they reflect internal ice sheet stratigraphy and deformation structures such as those reported here.

*Data availability.*  SEG-Y files of the 2D-focused SAR processed radar data are available from the UK Polar Data Centre https://data.bas.ac.uk/metadata.php?id=GB/NERC/BAS/PDC/00937. Ice thickness picks for the IMAFI survey are available from the UK Polar Airborne Geophysics Data Portal https://legacy.bas.ac.uk/data/aerogeo/dataset/imafi/.

*Author contributions.*  NR and MJS wrote the paper. HC and NR processed the data. NR collected and analyzed the data. MJS was PI of the IMAFI project. All authors contributed to the final version of the manuscript.

*Competing interests.*  The authors have no competing interests to declare.

*Acknowledgements.*  We thank: Institute and Möller ice stream Antarctic Funding Initiative (IMAFI) science and field team (Fausto Ferraccioli, Tom A Jordan, Rob Bingham, David Rippin, Anne Le Brocq, Carl Robinson, Ian Potten, Doug Cochrane, Mark Oosterlander); British Antarctic Survey logistics; John Woodward for discussions regarding entrainment of basal sediments; Rob Larter for information regarding

Antarctic tephra deposits dating to the last glacial cycle; Tom M Jordan for discussions on ice crystal fabric; TCD editor Joe MacGregor, Rob Bingham and four anonymous reviewers for helpful recommendations and comments on earlier versions of this manuscript. The IMAFI

project was funded by UK NERC AFI Grant NE/G013071/1. Aspects of this work were inspired/motivated by the Scientific Committee for Antarctic Research (SCAR) AntArchitecture community.

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

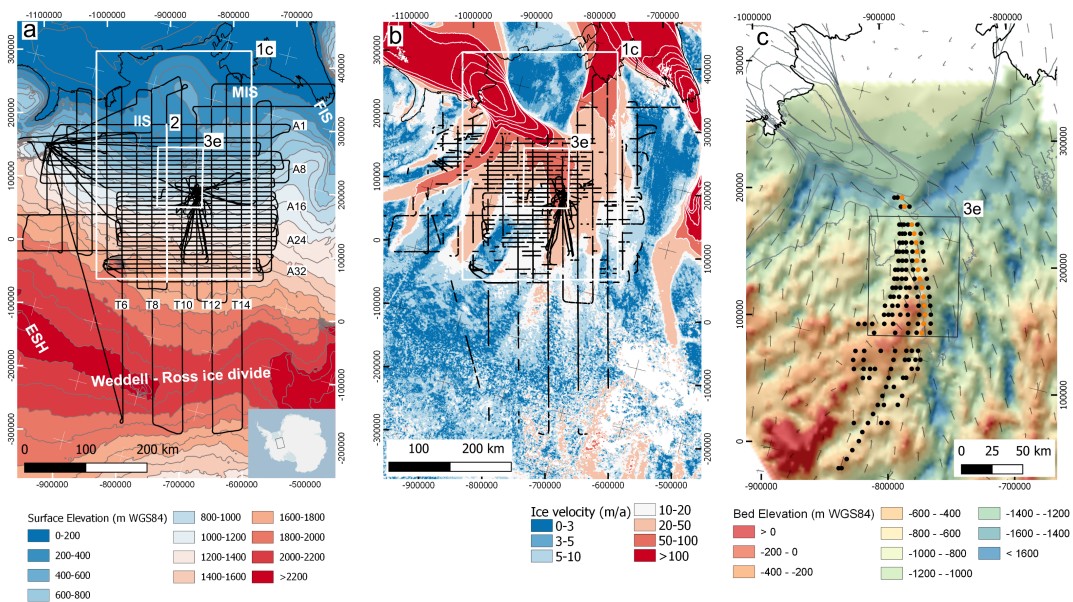

**Figure 1.** (a) Ice-penetrating radar survey flights over the catchments of the Institute and Möller ice streams (thick black lines), overlain on ice surface elevation (m WGS84) (Bamber et al., 2009). Thin black line in 1a-1c is MODIS grounding line (Bohlander and Scambos, 2007); Grey lines are ice surface elevation contours at 100 m intervals; Möller Ice Stream (MIS), Institute Ice Stream (IIS), Foundation Ice Stream (FIS), Weddell-Ross ice divide and Ellsworth Subglacial Highlands (ESH) are indicated. White boxes indicate extents of Figures 1c and 3e. White line is survey tie line 9, shown in figure 2. Inset shows Antarctic location of Figures 1a-b; (b) Observations of deep-ice unit R1 in radargrams across the study area (thick black lines), overlain on MEASURES ice velocity (Rignot et al., 2017). White lines are velocity contours at 50 ma$^{-1}$ intervals. Gaps in observations are where either data were not acquired or where the deep ice unit could not be identified in the radar data (e.g. because of signal attenuation etc.). White boxes indicate extents of Figures 1c and 3e; (c) Folded deep ice beneath IIS, relative to basal topography (Ross et al., 2012) and surface ice flow (Rignot et al. 2017). The location of fold axes peaks, identified in radargrams, are represented by black and orange circles, the latter colour denoting the position of the fold with the 'hand-shaped-reflection' (Figure 3d) at its core. Black box defines limits of Figure 3e. Thin grey lines are 50 m contours of MEASURES ice velocity (Rignot et al., 2017). Arrows show direction and magnitude of ice flow (Rignot et al., 2017).

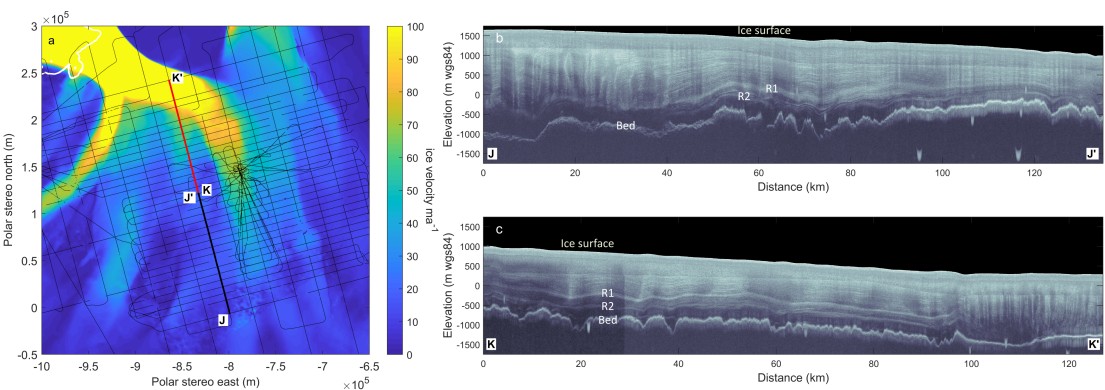

**Figure 2.** 260 km of radar data along tieline 9 demonstrating the typical stratigraphy of the ice column (high reflectivity meteoric ice, underlain by low reflectivity lower ice column with high reflectivity layers, and the brightness and widespread nature of the band of ice layers (R1 and R2) throughout the IIS and MIS catchments: (a) map of ice velocity (Mouginot et al., 2019) with IMAFI survey grid (thin black lines), grounding line (white line) and location of radar data in 2b (thick black line) and 2c (thick red line). Ice velocity colour scale is saturated at 100 ma$^{-1}$; (b) radargram of upper to mid IIS catchment (thick black line in 2a). Note bifurcation of R2 into 3-4 layers between 50-80 km; (c) continuation of radargram shown in b into the mid- to lower- IIS catchment (thick red line in 2a). Radargrams in 2b and 2c are radar survey 'tieline' 9, which is also shown in Figure 4 and Supplementary Figure 1. Isolated U-shaped 'blobs' are artefacts associated with SAR processing of the radar data.

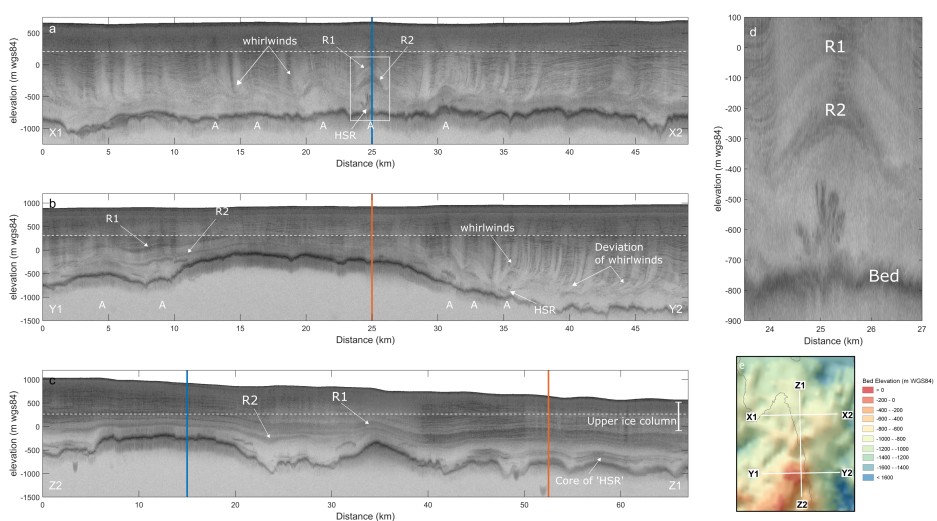

**Figure 3.** Ice-penetrating radargrams showing detailed englacial layering and folding: (a) Across-flow radargram (X1-X2), with englacial folds and hand-shaped reflection ('HSR'). View is down-ice. Intersection with along-flow radargram Z1-Z2 indicated by blue vertical line. Position of fold anticlines indicated ('A'). White box is location of Fig. 3d.; (b) Across-flow radargram (Y1-Y2), showing early development of englacial folds and their positions ('A'). Intersection with along-flow radargram Z1-Z2 indicated by red vertical line. (c) Down ice-flow radargram (Z1-Z2) connecting 3a and 3b. Ice flow is approximately left to right. Deep-ice units R1 and R2 are annotated. Intersections with radargrams X1-X2 and Y1-Y2 are indicated by blue and red vertical lines, respectively; (d) zoom-in of hand-shaped reflection from Fig 3a; (e) map of subglacial topography (Ross et al. 2012) showing positions (white lines) of radar transects X1-X2, Y1-Y2 and Z1-Z2. Thin grey line is 50 ma$^{-1}$ velocity contour (Rignot et al. 2018). Note that Fig. 3a has a different Y-axis scale to Figs. 3b and 3c.

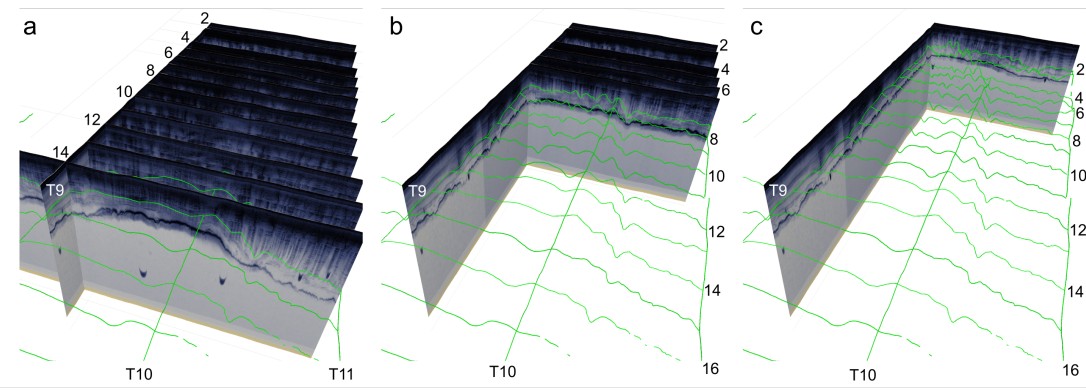

**Figure 4.** 3D visualisation of the picked (green lines) englacial layer R1 demonstrating 3D pattern of folding and density of survey lines: (a) radargram across ice flow lines 14-2; (b) radargram across ice flow line 7-2; (c) radargram across ice flow line 2. Ice flow is into page and there is 105 km between across flow lines 14 and 2. Long down-ice 'tie' line radargram shown in a-c is tie line 9 (figure 2c). Tie lines 9, 10 and 11 are annotated.

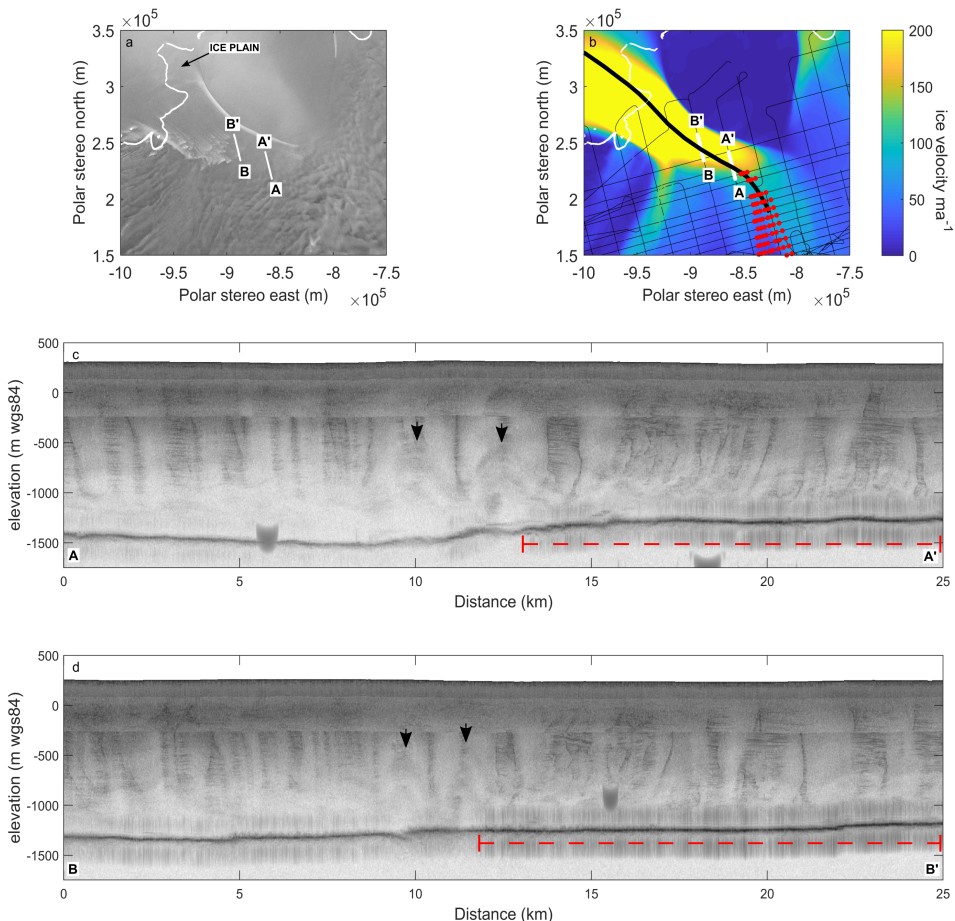

**Figure 5.** Englacial folds and ice flow of the IIS trunk: (a) RADARSAT image mosaic (Jezek et al., 2013) of the ice sheet surface of the IIS. Thick white lines show positions of radar data shown in 5c and 5d. Thin white line is grounding line (Rignot et al. 2011). The ice plain of the IIS is annotated; (b) mapped englacial fold axes (red dots) and prominent surface flow stripe (thick black line) mapped using RADARSAT and MODIS mosaics of Antarctica (Jezek et al., 2013; Haran et al., 2005), overlain on ice flow (Mouginot et al., 2019), IMAFI survey grid (thin black lines), and grounding line (thin white line) (Rignot et al. 2011). Thick white lines show positions of radar data shown in 5c and 5d; (c & d) radar data across the IIS trunk oblique to the flow stripe and ice flow, showing the subtle signature of the most apparent englacial folding (two black arrows) within the faster flowing trunk. The hand-shaped reflection (HSR) fold is the fold indicated by the right of two black arrows. A qualitative increase in basal reflectivity is apparent to the right of the HSR fold in both radargrams (indicated by dashed red line).

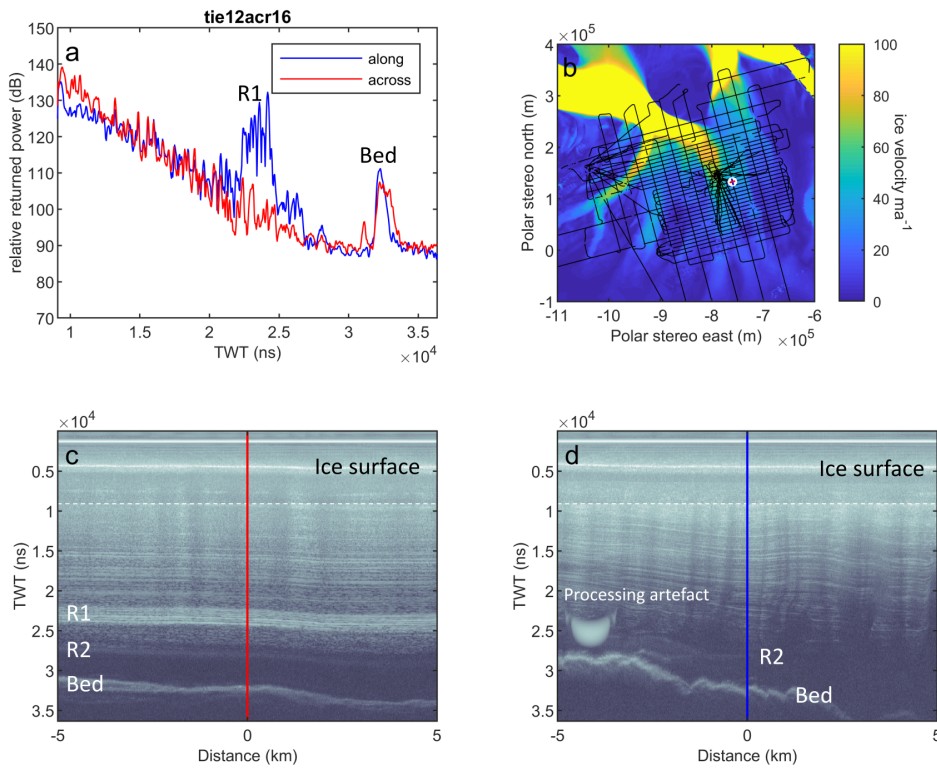

**Figure 6.** Example of the reflectivity of internal layering beneath Institute Ice Stream at cross over points: (a) relative returned power (dB) for eleven stacked 2D SAR processed traces (∼100 m of radar data) along (blue) and across (red) ice flow. Reflection packages R1 and the bed of the ice sheet are annotated; (b) location of radar data crossover (white filled circle with location of across -red- and along -blue- flow), underlain by radar survey grid (thin black lines) and ice velocity (Mouginot et al., 2019), with colour scale saturated at 100 ma$^{-1}$. (c) radar data along ice flow. Ice flow is left-to-right along radargram. Cross-over with across ice flow radar data (Fig. 6d) shown with red vertical line. Blue profile in 6a is the returned power of the crossover trace stacked with 5 traces either side of this 'red vertical line' cross over location; (d) radar data across ice flow. Ice flow is into radargram. Cross-over with along ice flow radar data (6c) shown with blue vertical line. Red profile in 6a is the returned power of the crossover trace stacked with 5 traces either side of this 'blue vertical line' cross over location. In distinct contrast to 6c, reflection package R1 is not imaged in these data. Along and across flow radar data (6c and 6d) were acquired with the same aircraft altitude, and therefore the same range to ice surface. This ensures that the geometry and anisotropy of the ice cannot be responsible for the pronounced anisotropy of the deep ice unit. White dashed lines in figures 6c and 6d define the boundary (200th vertical sample) between radar data SAR-processed (below line) and radar data not SAR-processed (above line). Information on SAR processing of the dataset is available in Jeofry et al., (2018a).