# Peer review of "Large-scale englacial folding and deep-ice stratigraphy within the West Antarctic Ice Sheet"

_The Cryosphere, 2019_

## Referee Comment (RC1) · Anonymous Referee #1 · 10 Dec 2019

General Comments:

This paper aims at explaining observations of englacial folding in the lower ice sheet column obtained from radio echo sounding. The deep-ice unit is analysed by mapping anticlines using a high-resolution grid of radio echo sounding data from 2010/11. Analysing the returned radar power at certain cross-overs allows to evaluate if englacial layers show signs of anisotropy. Evaluating different hypothesis explaining the formation of such complex near basal structures such as subglacial freeze-on, varying ice rheology and entrainment of basal material, the authors conclude that the deformation and folding of the near basal unit is related to convergent ice flow and layers of anisotropic ice.

The paper shows a nice and extensive data set of radargrams and returned radar

power at cross overs. However it is rather difficult to orient oneself between the different figures to really understand how the data is related to the ice flow. The different hypotheses are discussed but I feel that the evidence to choose one hypothesis over an other is lacking in parts. I'm not sure what the goal of the paper is - presenting the large-scale englacial folding or resolving how such a structure is obtained. I don't think the latter point is achieved.

Specific Comments:

- Line 4: this process might also involve freeze-on of basal water?

- Line 18: the citation of Dow et al. 2018 is not entirely correct in this context - as the impact of freeze-on units on ice-sheet flow and dynamics has not been investigated by that paper as they use a subglacial hydrology model and not an ice-flow model.

- Line 27-28: This sentence 'Like a structural geology problem, ...' is very assertive statement and I wonder why you are so sure about this. Does this come from the literature (but then references are missing) or does it come from your own findings (then say so)? Here in the introduction it seems to be at the wrong place.

- Line 57: You mention the ice thickness in this sentence but not how thick the lower-ice column is. It is mentioned in the abstract but here would be a good place too.

- Line 59: I think not only Figure 3 but also Figure 6 could be referred to.

- Line 60-61: R2 is referred to as a band of prominent reflection that "sometimes diverges and bifurcates, becoming a series of 3-4 layers". Is it the band that diverges and bifurcates or rather the series of layers within? I suggest a clearer wording.

- Line 64: I'm not sure if instead of Figure 2a you rather mean Figure 1d? It is not clear where exactly you refer to in Figure 2a - which tributary and wich lateral boundary (I guess in the center - parallel to K'K and J'J)?

- Line 67: Can you give examples where the amplitudes of up to 40% are seen? Maybe

highlight in one of the Figures so that it is clear to what you refer to.

- Line 69: For the statement that rather bed conformable undulations than folds are found along flow Figure 2b-c might not be the best example. Comparing the transects where the anticlines are marked (Figure 1d and 3a-b) it is obvious that the chosen along profile in 2 is not crossing many marked anticlines. Along flow one would expect not to see much of a fold that is oriented along flow (e.g. as shown in Figure 4 it is difficult to see the along flow pattern).

- Line 93-94: This surface flow stripe is not visible in any of the figures (neither main text nor supplement). It would be nice to see the mapped fold together with the flow stripe.

- Line 97: Could you highlight where the "significant shift in basal reflectivity" is seen? This would make it much clearer for the reader what you mean.

- Line 98: Same as above - highlight the onset zone of IIS so that it is very clear to what you are referring to.

-Lines 102-103: From the text/figures it is not clear to what exactly you are referring to. E.g.: - where is "grid-SW margin" to be found? - where is the ice plain? Mark/label it so that it is clear. I can guess but rather would like to be sure. Add these explanations to the overview Figure 1.

- Line 113: I'm not sure that the observation of no high relief in basal topography is enough to say that anisotropy is not forming in soft layers due to enhanced stresses on the stoss-faceof basal hills. One can imagine that if locally basal freeze-on exists, then the plume of accreted ice pushing into the ice columne might act as an obstacle to the meteoric ice flow.

- Lines 133-134: The usage of deep-ice units in this sentence is not clear as it seems to be used in two different contexts: 1) as the basal freeze-on units of accreted ice from the bed raising into the ice sheet as hypothesised by Bell et al, 2011; and 2) as the

unit of ice between R1 and the bed. So I wonder what in this paper is understood by the "deep-ice units". It is defined as the ice between R1 and the bed. I don't think you expect the whole unit to be frozen on by basal water. With basal freeze-on one would expect the to have some local areas where "material" is rising into the ice sheet and advecting downstream - shaping the meteoric layers above.

- Lines 134-140: The arguments against basal freeze-on, of slow-flowing and thin ice likely frozen to the bed, as well as fast-flowing ice over a wet bed, do not entirely exclude the formation of basal freeze-on plumes. In order to exclude basal freeze-on the possibility of water needs to be excluded. Where is the evidence that there is no water at the base of the ice-sheet? Is it visible in the radar data? It would make sense to map the anticlines together with the main topographically-constrained subglacial drainage network to evaluate if basal freeze-on is possible. Further the ice velocity itself cannot exclude the formation/existence of substantial freeze-on units.

- Lines 147-149: I don't understand the argument that because R1 and R2 have a consistent stratigraphic position "the structures and extent of the deep-ice structures must be the result of the deformation and localized folding of meteoric ice". How is this meant - a stratigraphic position in the vertical ice column? They will always be on top of each other - unless one is melted away. E.g. basal freeze-on would lead to the same result just pusing up the entire ice column - with the vertical distance between the layers diminishing.

- Line 148: What exactly is your deep ice structure? In parts it sounds that it is all the ice below R1? Or do you mean the layer pattern at depth? Is there a difference between units and structure?

- Lines 149-151: It is not clear to me how the anisotropy observed in a layer leads to the folding of the ice unit below. In the paper by Bons et al., 2016 it is not one anisotropic layer that leads to folding but a body of anisotropic ice.

- Lines 169-171: Why would the spiraling flow of basal ice only happen at that specific

location and why only on one side of the ice stream onset?

- Lines 173-176: "strongly contrasting physical and rheological properties of glacial and interglacial ice" might not be the only explanation that basal units are" folded, sheared and overturned".

- Line 203: I do not agree that you "demonstrated" that the deep-ice units have different physical properties than the ice above. You showed that one band does show in some places anisotropic behaviour.

- Line 204-206: I'm not happy with the statement in this sentence. The paper does not show that convergent flow heavily deforms deep-ice units, that this process leads to the formation of large-scale englacial folds and that these folds modulate ice-stream position structure and dynamics.

- Conclusions: The conclusion seems to me in parts too assertive. In my view the study is not really conclusive how this structures are really formed. I agree not so much with the top part but agree with the bottom half. It is still not clear what exactly is meant by deep-ice unit.

- Figures 1: The figures in the paper are not very well related to each other. It is difficult to see where the transects are taken relative to ice velocity, bed topography surface slope and the mapped anticlines. I suggest that the boxed area of 1d is not only shown in 1a but everywhere in 1. Ideally 1d would show bedtopography upstream of the section in 1d. Help reader to orientate by defining the orientation of the grid used in the figure.

- Figure 2: Where is this line (2a) in Figure 1 - maybe useful to show the tie line 1 and the across line 1 (or other number) as to allow reader to orient oneself in the overview figure. An overview of the shown radar profiles would be nice (see in Figure 1).

- Figure 3: What is the criteria for a fold - in 3b I can't see a fold in the radargram where the third "A" from left (from Y1 towards Y2) is. Further comparing the mapped "A" with

Figure 1d it seems to me that sth. is wrong. In 1d "HSR" is the second from right and not the third as shown in radargram and the spacing between the folds is different than in overview Figure 1d.

- Figure 4: Numbering the radar lines would be helpful (eventhough one can deduce the numbering) especially in context with Figure 1 (if there is some numbering).

- Figure 5: Distance taken from where for Figures 5b and c? Why not just show the "length" of these two radargrams (as a distance is o.k.). Why is the "signature of englacial folding" (caption line 5) not mapped/marked? What is the criteria for mapping these folds?

- Figure 6: in (b) why not mark in the white circle the along line in blue and across in red. In (c) and (d) it's not entirely clear if the colour of the vertical line represents the colour of the crossing transect. Maybe it makes more sense to mark in (c) in red where the across transect crosses and vice versa in (d). The caption explaining this is slightly confusing. For (c) and (d) it's not clear where -5 and 5 is e.g. is (c) going downstream from -5 to 5 or upstream? Are the across profiles (d) always oriented the same way? An orientation would help.

- Figures in Supplement: I'm not sure if the across profiles all (d)'s are always along the same orientation. Especially when looking at "HSR" in along tie10 it seems to me that it "jumps" from one across profile to the other. It would be nice to have the same geographical orientation for all along and across profiles at all times. As mentioned above it would be nice to know where the tie lines are.

Technical corrections:

- Caption 2: Line 2 - "ubiquity and widespread" is this not the same message? - Caption 2: Line 2 - band of ice-layers rather than "deep-ice layers" - Caption 2: Line 4 - "thin black line" of grounding line is not easy to differentiate from the "thin black lines" of the survey grid. - Caption 2: Line 5 - "bifurcation of R2 into 3-4 layers" happens also earlier

around 50 km.  - Caption 2: Line 5 - in (c) mention tie line 9 as used in Figure 4 and supplementary figures. - Caption 2: Line 6 - replace "show" with "shown".

- Caption 5: Line 4 - the transects are not "perpendicular" to the ice flow rather at an angle of 45 degrees oblique to ice flow.

---

## Referee Comment (RC2) · Anonymous Referee #2 · 3 Jan 2020

Review "Large-scale englacial folding and deep-ice stratigraphy within the West Antarctic Ice Sheet" by Neil Ross et al.

This paper presents an extensive data set of airborne radar across the tributaries of the Institute Ice Stream. The high-quality data makes it possible to track folds in the lower part of the ice column and investigate the nature of a particular reflector band with a directional dependency of reflection strength, indicating crystal anisotropy in the lower layers of ice. Different ice rheology due to anisotropy and the redistribution of this ice in the folding process is identified as playing a role for the organization of inflow to the ice stream. This is an interesting data set, with most profiles published in the supplements, and highlights the importance of considering ice rheology and anisotropy when trying to understand large scale ice flow.

[Figure]

Technically the manuscript is sound and well written. But I would like to raise some points which could be improved in the final version. My main concern is how the anisotropy of the ice crystal fabric is discussed. It is known from ice core data, seismic studies and also from numerical modelling that anisotropic ice is to be expected in ice sheets, and that this is linked to the dynamic setting and the deformation the ice has been subject to, as well as the influence of impurities in the ice. This is briefly mentioned here. However, there are different types of anisotropy, which are again linked to certain deformation regimes. To me it is not clear from this manuscript what kind of change in crystal fabrics is causing the reflection package described. The directional dependency of the reflector strength would indicate that it is a girdle type, typical for extensional flow, as a single maximum distribution (typical for shear) is symmetric to the vertical and therefore would not be different in profiles at different angles. I think it would be essential to discuss the different types of anisotropy and how they are linked to dynamics, to be able to interpret the influence on ice stream flow of the anisotropic ice package. As it stands now I would not agree that the data support the strong conclusion that is has been shown how anisotropy in the lower layers "modulates ice stream position, structure and dynamics). I do not disagree in general, I just think the authors have to be more convincing in their line of arguments. In summary I think there is need for a better interpretation of the links between anisotropy, ice dynamic setting, and folding structures.

Some smaller points: Lines 27-30: Folds can be generated in an anisotropic material by lateral compression, there is no need for a rheology contrast between two layers. Buckle folding, when a hard layer is embedded in a softer matrix, produces different kinds of folds (parallel folds), and the rheology contrast needed for this is much bigger than the range expected in natural ice. The paragraph reads as if the contrast in rheology is the origin of folding.

Lines 103-104: to interpret the fold here as a natural boundary for the stream is a bit circular. The folding is linked to dynamics, so the fold where the dynamic setting is right

(shear margin).

Line 109, Comment on the supplementary figures: It is great to be able to see all the data, but would be really helpful would be a way to quantify the difference between the two profile directions in the relevant depth range and then plot this color-coded on a map.

Lines 112-113: Maybe this is the place to describe what kind of anisotropy you would expect and why.

Line 150: Do you mean a band of ice, that is in the lower part the ice should be isotropic again? Or is it a thick basal layer?

Line 155: As mentioned above, how can you conclude that the fold is influencing the location of the shear margin? In the shear margin there is in general a compressive stress across flow, so it would be the other way around.

Line 165: I don't think that it would be possible to form fractures at the base of an ice sheet. The publication which is cited is about a mountain glacier, only a few hundreds of meters thick. This must be some other form of ductile entrainment of bottom material.

---

## Author Comment (AC1) · 21 Jan 2020

**Response to Reviewer 1 for: "Large-scale englacial folding and deep-ice stratigraphy within the West Antarctic Ice Sheet" (MS No: tc-2019-245)**

We are grateful to both reviewers for their constructive and helpful reviews of our manuscript. Below we respond (non-highlighted text) to the comments of reviewer 1 (highlighted in grey).

**Anonymous Referee #1**
*General Comments:*

*This paper aims at explaining observations of englacial folding in the lower ice sheet column obtained from radio echo sounding. The deep-ice unit is analysed by mapping anticlines using a high-resolution grid of radio echo sounding data from 2010/11. Analysing the returned radar power at certain cross-overs allows to evaluate if englacial layers show signs of anisotropy. Evaluating different hypothesis explaining the formation of such complex near basal structures such as subglacial freeze-on, varying ice rheology and entrainment of basal material, the authors conclude that the deformation and folding of the near basal unit is related to convergent ice flow and layers of anisotropic ice. The paper shows a nice and extensive data set of radargrams and returned radar power at cross overs. However it is rather difficult to orient oneself between the different figures to really understand how the data is related to the ice flow. The different hypotheses are discussed but I feel that the evidence to choose one hypothesis over an other is lacking in parts. I'm not sure what the goal of the paper is - presenting the large-scale englacial folding or resolving how such a structure is obtained. I don't think the latter point is achieved.*

We thank reviewer 1 for their review and helpful suggestions to improve the manuscript. We very much appreciate the description of our "nice and extensive dataset of radargrams". We believe that this dataset is unique for a modern SAR-processed radio-echo sounding survey in Antarctica. There are certainly few airborne RES surveys of an appropriate orientation and such closely spaced survey lines across the onset zone of a major Antarctic ice stream system like the Institute-Möller ice stream complex.

We would disagree with the comment that it is "…rather difficult to orient oneself between the different figures to really understand how the data is related to the ice flow…", as we have provided maps of ice flow in four of the six figures, whilst the location of figure 3d (figure 3 is one of the two figures that does not have its own ice flow map) is shown on figure 1d, where ice flow contours are shown above a map of bed elevation. However, we are happy to adopt many of the later specific suggestions from reviewer 1 for our figures (i.e. labelling tie and across lines, adding the location of radargrams in figure 2 to figure 1a, and having additional area boxes in figure 1). We hope that accepting these should also address reviewer 1's general comment.

Reviewer 1 states that they are "not sure what the goal of the paper is - presenting the large-scale englacial folding or resolving how such a structure is obtained". The primary purpose of the manuscript is to report the discovery of the large-scale englacial folding AND the unusual deep ice stratigraphy (i.e. extensive thick high reflectivity layers at depth) across the study area. Secondary goals of the manuscript are to infer the physical properties of these folds and layers; to develop hypotheses for how the layers, and associated structures, might form; and to assess any potential glaciological impact. Whilst our "nice and extensive dataset" can be used to identify the englacial structures, and, in combination with ice velocity datasets, can be used to describe its spatial relationship with ice flow, we acknowledge that it does have its limitations. Whilst we can infer some physical properties (i.e. strong preferred crystal fabric) from the data, we lack direct measurements of physical properties (e.g. from ice cores corresponding to the airborne RES survey) to 'calibrate' the radar data, so we are unable, at present, to resolve "how such a structure is obtained". However, from the available datasets, and with reference to glaciological theory, combined with existing observations

elsewhere in Antarctica and Greenland, we have been able to develop initial explanations for "resolving how such a structure is obtained". Full determination of the processes responsible for the englacial structures is beyond the current dataset and therefore this paper, however, and will need to await future direct investigations and observations (e.g. coring and geophysical measurements) designed specifically for addressing this question. We note that the current RES dataset was not designed, or acquired, to answer such a question; it was designed to test for the presence or absence of subglacial sediments beneath Institute Ice Stream, and the discovery of the englacial structures and layers was serendipitous. Given this context, we would argue that we have successfully reported the englacial layering and structures, inferred possible physical properties, proposed possible processes responsible for fold formation, and assessed the glaciological implications of the folding. That we have not fully resolved "how such a structure is obtained" is not, in our opinion, a major issue. We have undertaken the best assessment and interpretation of the evidence possible, given the data currently at our disposal.

*Specific Comments:*
*- Line 4: this process might also involve freeze-on of basal water?*
We acknowledge that local freeze-on of basal water is possible, but the widespread and extensive nature of the deep-ice facies we report is inconsistent with the freeze-on of basal water being an important process in its formation. The sentence in line 4 reports that we observe little evidence for basal water within our study area (e.g. there are few definite subglacial lakes) as detailed later in the manuscript. We therefore propose no change here. Additional discussion of this issue is outlined in our comments on lines 134-140 below.

*- Line 18: the citation of Dow et al. 2018 is not entirely correct in this context - as the impact of freeze-on units on ice-sheet flow and dynamics has not been investigated by that paper as they use a subglacial hydrology model and not an ice-flow model.*
We propose that we simply remove the reference to this paper at this location (though we retain it elsewhere in the manuscript).

*- Line 27-28: This sentence 'Like a structural geology problem, ...' is very assertive statement and I wonder why you are so sure about this. Does this come from the literature (but then references are missing) or does it come from your own findings (then say so)? Here in the introduction it seems to be at the wrong place.*
We propose inserting references to papers by MacGregor and others, JGR, 2015 https://agupubs.onlinelibrary.wiley.com/doi/10.1002/2014JF003215 and Hudleston, Journal of Structural Geology, 2015 https://www.sciencedirect.com/science/article/pii/S0191814115300365 here. Both these papers draw attention to similarities between ice sheet englacial structures and structural geology, and their inclusion should hopefully address reviewer 1'a comment.

*- Line 57: You mention the ice thickness in this sentence but not how thick the lower-ice column is. It is mentioned in the abstract but here would be a good place too.*
We will insert a value or range for the thickness of the lower ice column here.

*- Line 59: I think not only Figure 3 but also Figure 6 could be referred to.*
We agree. We will add the reference to figure 6 here.

*- Line 60-61: R2 is referred to as a band of prominent reflection that "sometimes diverges and bifurcates, becoming a series of 3-4 layers". Is it the band that diverges and bifurcates or rather the series of layers within? I suggest a clearer wording.*

We propose rewording lines 59-61 to: "Beneath R1, within the low-reflectivity lower-ice column, a second highly reflective englacial reflection (R2) is observed. R2 sometimes diverges and bifurcates, becoming a series of 3-4 layers (Figure 2b)."

*- Line 64: I'm not sure if instead of Figure 2a you rather mean Figure 1d? It is not clear where exactly you refer to in Figure 2a - which tributary and wich lateral boundary (I guess in the center - parallel to K'K and J'J)?*
We will refer to figure 1d instead of 2a.

*- Line 67: Can you give examples where the amplitudes of up to 40% are seen? Maybe highlight in one of the Figures so that it is clear to what you refer to.*
We will annotate this on figure 5a.

*- Line 69: For the statement that rather bed conformable undulations than folds are found along flow Figure 2b-c might not be the best example. Comparing the transects where the anticlines are marked (Figure 1d and 3a-b) it is obvious that the chosen along profile in 2 is not crossing many marked anticlines. Along flow one would expect not to see much of a fold that is oriented along flow (e.g. as shown in Figure 4 it is difficult to see the along flow pattern).*
The purpose of referring to 2b-c here was that these radargrams demonstrate that there are no folds or overturnings in radar data acquired in that orientation. They were presented as typical examples of the survey data and show that the deep reflective layers are very extensive (i.e. over distances of ~250 km or more) and bed conformable. If referring to figures 2b-c in line 69 is an issue we can simply remove reference to 2b-c, as 3c demonstrates the same point just not over such a large spatial area. However, we would prefer to leave line 69 as is. The evidence for bed conformable layering down flow is inconsistent with localised basal freeze-on being behind the formation of the deep ice units. Such a process could not explain this layer stratigraphy and structure.

*- Line 93-94: This surface flow stripe is not visible in any of the figures (neither main text nor supplement). It would be nice to see the mapped fold together with the flow stripe.*
The position of the surface flow stripe is mapped in figure 5a, but we propose adding an additional subplot (comparable to size and area of 5a) to figure 5 that shows surface imagery of ice stream trunk (e.g. RADARSAT) with the flow stripe visible. This will provide the 'raw data' of the surface flow stripe requested by reviewer 1.

*- Line 97: Could you highlight where the "significant shift in basal reflectivity" is seen? This would make it much clearer for the reader what you mean.*
We will mark/annotate 'bright' and 'dim' areas of the ice sheet bed on the radargrams in figure 5.

*- Line 98: Same as above - highlight the onset zone of IIS so that it is very clear to what you are referring to.*
We will annotate figure 5a with this information.

*-Lines 102-103: From the text/figures it is not clear to what exactly you are referring to. E.g.: - where is "grid-SW margin" to be found? - where is the ice plain? Mark/label it so that it is clear. I can guess but rather would like to be sure. Add these explanations to the overview Figure 1.*
The ice plain is currently located just beyond the grid-north limit of 5a. We will expand the area of figure 5a to include it. We will also annotate/map the ice plain on the figure.

*- Line 113: I'm not sure that the observation of no high relief in basal topography is enough to say that anisotropy is not forming in soft layers due to enhanced stresses on the stoss-face of basal hills. One*

*can imagine that if locally basal freeze-on exists, then the plume of accreted ice pushing into the ice columne might act as an obstacle to the meteoric ice flow.*

We propose to simply delete the sentence at the end of the paragraph that dismisses this idea (i.e. lines 112-113), and that starts "We do not observe…..".

*- Lines 133-134: The usage of deep-ice units in this sentence is not clear as it seems to be used in two different contexts: 1) as the basal freeze-on units of accreted ice from the bed raising into the ice sheet as hypothesised by Bell et al, 2011; and 2) as the unit of ice between R1 and the bed. So I wonder what in this paper is understood by the "deep-ice units". It is defined as the ice between R1 and the bed. I don't think you expect the whole unit to be frozen on by basal water. With basal freeze-on one would expect the to have some local areas where "material" is rising into the ice sheet and advecting downstream - shaping the meteoric layers above.*

We confirm that for our study area, the "deep-ice unit" is the body of ice between R1 and the bed. To limit confusion in lines 133-134, we propose to substitute "deep-ice unit" with "basal ice" when referring to the paper on basal ice features in East Antarctica by Bell et al. 2011. The sentence would then read: "The 'freeze-on' hypothesis for the formation of basal ice (Bell et al., 2011) cannot explain the deep-ice unit and incorporated layers (i.e. R1 to the ice sheet bed) for four reasons:…….".

*- Lines 134-140: The arguments against basal freeze-on, of slow-flowing and thin ice likely frozen to the bed, as well as fast-flowing ice over a wet bed, do not entirely exclude the formation of basal freeze-on plumes. In order to exclude basal freeze-on the possibility of water needs to be excluded. Where is the evidence that there is no water at the base of the ice-sheet? Is it visible in the radar data? It would make sense to map the anticlines together with the main topographically-constrained subglacial drainage network to evaluate if basal freeze-on is possible. Further the ice velocity itself cannot exclude the formation/existence of substantial freeze-on units.*

We can include an additional figure of the mapped anticlines with the subglacial hydrological drainage network in the manuscript if required. However, even if we did this it could not explain or dismiss the widespread presence of the strongly reflective deep ice layers across the study area (see lines 52-55 of the manuscript), even in areas that are clearly underlain by cold based ice.

With regards to the questions "Where is the evidence that there is no water at the base of the ice-sheet?" and "Is it visible in the radar data?" we refer to lines 139-146 of the manuscript, where we describe the limited evidence (in our, and other, RES data) for "significant ponding of subglacial water" in the study area.

*- Lines 147-149: I don't understand the argument that because R1 and R2 have a consistent stratigraphic position "the structures and extent of the deep-ice structures must be the result of the deformation and localized folding of meteoric ice". How is this meant - a stratigraphic position in the vertical ice column? They will always be on top of each other - unless one is melted away. E.g. basal freeze-on would lead to the same result just pusing up the entire ice column - with the vertical distance between the layers diminishing.*

It is true that basal freeze-on would lead to the same result by pushing up the entire ice column, but it is physically unrealistic for this to have occurred over such a widespread area as we observe R1 and R2. The volumes of water required would be, quite simply, enormous and freeze-on rates would have to be consistent right across the study area. We might expect freeze-on locally (and we do not dispute the possibility of localised basal freeze-on), but not over the extensive area that we map the layers (see Lines 52-55 and Figure 1 – noting that there is an error in the mapping of R2 in figure 1b which significantly under-reports its spatial distribution. We will rectify this in any revised version of the manuscript). In addition, freeze-on cannot be responsible for the formation of the basal ice layers over high subglacial mountain ranges, e.g. the Ellsworth Subglacial Highlands, where the ice is cold based and has likely been for millions of years. Therefore, the only physically realistic explanation for the

consistent stratigraphic position of R1 and R2 in the ice column, over such a vast geographical area, is that they reflect layers that were initiated at the surface by meteoric deposition (i.e. of snow and/or volcanic ash), but that will have undergone physical modification (i.e. to crystal fabric orientation) in the ice column during burial and strain. We do propose that figure 1 (in addition to figure 2) is referenced in line 147 however, to direct the reader to the extensive geographical distribution of the deep ice layers.

*- Line 148: What exactly is your deep ice structure? In parts it sounds that it is all the ice below R1? Or do you mean the layer pattern at depth? Is there a difference between units and structure?*
By deep ice we mean all the ice between R1 and the bed. Units are the layers of ice (like geological units), the deep ice structure refers to the geometry of the units (e.g. folds) in the deep ice.

*- Lines 149-151: It is not clear to me how the anisotropy observed in a layer leads to the folding of the ice unit below. In the paper by Bons et al., 2016 it is not one anisotropic layer that leads to folding but a body of anisotropic ice.*
In response to a comment from reviewer 2, we propose to amend these lines to: "Further, given the radar anisotropy observed, the most likely explanation for the folds is that they are caused by a combination of convergent ice flow and the distinct physical (i.e. crystal orientation fabric), and consequent rheological, properties of the band of ice associated with R1.". As is clear from figure 6 (and multiple other examples in the supplementary information) R1 is a broad band (described in line 58 as "up to 200 m" thick) that is rather different from the narrow reflections we typically think of as englacial layers observed in RES data. As such, R1 is more a body or package of ice than a single layer (and therefore potentially like Bons et al, 2016). Furthermore, we do not know the properties of most of the ice below R1 as, unlike in Greenland, as we do not have direct observations (e.g. from ice cores) to calibrate against. We can infer from the RES data that R1 is anisotropic, and that R2 is likely isotropic. However, we do not know the physical properties of the rest of the deep ice units as it is typically an echo-free zone below R1. It may be anisotropic, either in places or throughout, but we cannot determine that from the currently available data.

*- Lines 169-171: Why would the spiraling flow of basal ice only happen at that specific location and why only on one side of the ice stream onset?*
We propose the deletion of lines 169-171 and the reference to Schoof and Clarke 2008.

*- Lines 173-176: "strongly contrasting physical and rheological properties of glacial and interglacial ice" might not be the only explanation that basal units are" folded, sheared and overturned".*
We accept that englacial layers in Greenland and Antarctica can be deformed by the freeze-on of basal ice (which we assume reviewer 1 is alluding to here), but based on the currently available literature for Greenland (which we cite in lines 175-176) the most widespread accepted explanation for such folding is the strong contrast in physical and rheological properties between glacial ('soft') and interglacial ('hard') ice. We do not see the need to alter this sentence.

*- Line 203: I do not agree that you "demonstrated" that the deep-ice units have different physical properties than the ice above. You showed that one band does show in some places anisotropic behaviour.*
We propose rewording this sentence to: "We have demonstrated the presence of an extensive package of deep-ice units beneath the Institute and Moller ice streams. At least one layer in the deep ice has physical properties (i.e. ice crystal orientation fabric, rheology) significantly different to the upper ice column."

*- Line 204-206: I'm not happy with the statement in this sentence. The paper does not show that convergent flow heavily deforms deep-ice units, that this process leads to the formation of large-scale englacial folds and that these folds modulate ice-stream position structure and dynamics.*

We propose rewording this sentence to: "At the lateral boundary of the onset of enhanced ice flow of IIS, where ice flow is convergent, these deep-ice units have been heavily deformed. Deformation has led to the development of large-scale englacial folds that may modulate ice-stream position, structure and dynamics."

*- Conclusions: The conclusion seems to me in parts too assertive. In my view the study is not really conclusive how this structures are really formed. I agree not so much with the top part but agree with the bottom half. It is still not clear what exactly is meant by deep-ice unit.*

We suggest that we have addressed this issue with changes we propose to lines 203, and lines 204-206 (see above) that are less "assertive" than they were. We acknowledge that our study may not be conclusive as to how the structures formed, but that reflects the serendipitous nature of the discovery and the datasets currently available. More conclusive understanding will require additional work (i.e. new geophysical/ice core observations and/or numerical modelling) beyond this 'discovery' paper. As stated previously, the deep ice unit is all the ice between R1 and the bed.

*- Figures 1: The figures in the paper are not very well related to each other. It is difficult to see where the transects are taken relative to ice velocity, bed topography surface slope and the mapped anticlines. I suggest that the boxed area of 1d is not only shown in 1a but everywhere in 1. Ideally 1d would show bedtopography upstream of the section in 1d. Help reader to orientate by defining the orientation of the grid used in the figure.*

We can certainly add the boxed area shown in 1d in 1a-c. We are also happy to expand the extent of 1d up-ice. We are unclear what the reviewer means by "*Help reader to orientate by defining the orientation of the grid used in the figure.*". The orientation of the 1a-d should be clear from the inset figure in 1a, but we are happy to modify the figure in response to a specific suggestion as to how to address this comment.

*- Figure 2: Where is this line (2a) in Figure 1 - maybe useful to show the tie line 1 and the across line 1 (or other number) as to allow reader to orient oneself in the overview figure. An overview of the shown radar profiles would be nice (see in Figure 1).*

We propose to add the location of the radar data shown in 2b-c to figure 1a as well as showing it in 2a.

*- Figure 3: What is the criteria for a fold - in 3b I can't see a fold in the radargram where the third "A" from left (from Y1 towards Y2) is. Further comparing the mapped "A" with Figure 1d it seems to me that sth. is wrong. In 1d "HSR" is the second from right and not the third as shown in radargram and the spacing between the folds is different than in overview Figure 1d.*

Folds were mapped where the englacial layers were folded and were not bed conformable. In the case of 3b, we may have been slightly overenthusiastic in our identification of the third A from the left, based on our mapping of the fold in parallel radar lines (i.e. we tracked it back to that location on the radargram from down-ice radargrams). We will review all our picks of englacial layer folds to ensure that only folds that we can be 100% confident represent folds are mapped. With regards to the 'HSR' we have clearly made an error in the manual labelling of figure 3b or the plotting of the fold axes in 1d. Further up-ice (i.e. where 3b is located), it does become harder to confidently identify the folds. We will address the discrepancy in any revised version of the manuscript after a comprehensive review of all the picks of the englacial folds.

*- Figure 4: Numbering the radar lines would be helpful (eventhough one can deduce the numbering) especially in context with Figure 1 (if there is some numbering).*

We can add numbering (perhaps every second or third across line) to figures 1 and 4 to aid with this.

*- Figure 5: Distance taken from where for Figures 5b and c? Why not just show the "length" of these two radargrams (as a distance is o.k.). Why is the "signature of englacial folding" (caption line 5) not mapped/marked? What is the criteria for mapping these folds?*
The distance is from the start of the radargram. This is an error and can be corrected. We also need to amend Figure 3 x-axes so that distance is reported in km rather than m. We can add annotation to the englacial folds that are apparent in figure 5. Mapping of the folds is based on obvious deviation of englacial layering from the bed profile, and identification of folding in up-ice radargrams.

*- Figure 6: in (b) why not mark in the white circle the along line in blue and across in red. In (c) and (d) it's not entirely clear if the colour of the vertical line represents the colour of the crossing transect. Maybe it makes more sense to mark in (c) in red where the across transect crosses and vice versa in (d). The caption explaining this is slightly confusing. For (c) and (d) it's not clear where -5 and 5 is e.g. is (c) going downstream from -5 to 5 or upstream? Are the across profiles (d) always oriented the same way? An orientation would help.*
Because of the scale of the figure we were concerned that two 10 km-long plots of the radargram locations would be lost, hence why we opted for the circle and cross to demonstrate location. We will explore options for plotting the XY positions of the radar traces over the white circles to address the reviewer's comment.
We are happy to swop the colours of the vertical lines in the radargrams. This makes sense.
The radargrams in c and d are oriented along the direction of the survey flight, so are not always oriented the same way as different parallel survey flights will have been flown in different directions. We can either: (a) orient all the radargrams in the same way; or (b) leave them as is, and simply label the orientation.

*- Figures in Supplement: I'm not sure if the across profiles all (d)'s are always along the same orientation. Especially when looking at "HSR" in along tie10 it seems to me that it "jumps" from one across profile to the other. It would be nice to have the same geographical orientation for all along and across profiles at all times. As mentioned above it would be nice to know where the tie lines are.*
No, the across profiles are not always in the same orientation, as they are oriented in the direction of the survey flight, and different survey flights were flown in different directions. As stated in the previous response we can either: (a) orient all the radargrams in the same way; or (b) leave them as is, and simply label the orientation. From this comment, it appears that reviewer 1 would prefer the former of these two options.
Tie and across lines will be labelled on figure 1 and 4 (see response to comment on figure 4).

*Technical corrections:*
*- Caption 2: Line 2 - "ubiquity and widespread" is this not the same message?*
We propose deleting "ubiquity"

*- Caption 2: Line 2 - band of ice-layers rather than "deep-ice layers"*
We are happy to make this change.

*- Caption 2: Line 4 - "thin black line" of grounding line is not easy to differentiate from the "thin black lines" of the survey grid.*
Agreed, we will change the colour of the grounding line.

*- Caption 2: Line 5 - "bifurcation of R2 into 3-4 layers" happens also earlier around 50 km.*
Agreed. We had not spotted this. We will amend the caption accordingly.

*- Caption 2: Line 5 - in (c) mention tie line 9 as used in Figure 4 and supplementary figures.*
Agreed. We will mention tie line 9.

*- Caption 2: Line 6 - replace "show" with "shown".*
Agreed. We will amend the caption accordingly

*- Caption 5: Line 4 - the transects are not "perpendicular" to the ice flow rather at an angle of 45 degrees oblique to ice flow.*
Agreed. We will amend the caption accordingly.

---

## Author Comment (AC2) · 21 Jan 2020

**Response to Reviewer 2 for: "Large-scale englacial folding and deep-ice stratigraphy within the West Antarctic Ice Sheet" (MS No: tc-2019-245)**

We are grateful to both reviewers for their constructive and helpful reviews of our manuscript. Below we respond (non-highlighted text) to the comments of reviewer 2 (highlighted in grey).

**Anonymous Referee #2**

*This paper presents an extensive data set of airborne radar across the tributaries of the Institute Ice Stream. The high-quality data makes it possible to track folds in the lower part of the ice column and investigate the nature of a particular reflector band with a directional dependency of reflection strength, indicating crystal anisotropy in the lower layers of ice. Different ice rheology due to anisotropy and the redistribution of this ice in the folding process is identified as playing a role for the organization of inflow to the ice stream. This is an interesting data set, with most profiles published in the supplements, and highlights the importance of considering ice rheology and anisotropy when trying to understand large scale ice flow.*

*Technically the manuscript is sound and well written. But I would like to raise some points which could be improved in the final version.*

We thank reviewer 2 for their review and helpful suggestions to improve the manuscript. We very much appreciate the recognition that our data is "high-quality", "interesting" and "highlights the importance of considering ice rheology and anisotropy when trying to understand large scale ice flow". We are also delighted that they find that "Technically the manuscript is sound and well written".

*Technically the manuscript is sound and well written. But I would like to raise some points which could be improved in the final version.* ***My main concern is how the anisotropy of the ice crystal fabric is discussed.*** *It is known from ice core data, seismic studies and also from numerical modelling that anisotropic ice is to be expected in ice sheets, and that this is linked to the dynamic setting and the deformation the ice has been subject to, as well as the influence of impurities in the ice. This is briefly mentioned here. However, there are different types of anisotropy, which are again linked to certain deformation regimes. To me it is not clear from this manuscript what kind of change in crystal fabrics is causing the reflection package described. The directional dependency of the reflector strength would indicate that it is a girdle type, typical for extensional flow, as a single maximum distribution (typical for shear) is symmetric to the vertical and therefore would not be different in profiles at different angles. I think it would be essential to discuss the different types of anisotropy and how they are linked to dynamics, to be able to interpret the influence on ice stream flow of the anisotropic ice package.*

*As it stands now I would not agree that the data support the strong conclusion that is has been shown how anisotropy in the lower layers "modulates ice stream position, structure and dynamics). I do not disagree in general, I just think the authors have to be more convincing in their line of arguments. In summary I think there is need for a better interpretation of the links between anisotropy, ice dynamic setting, and folding structures.*

We are very grateful to reviewer 2 for highlighting this issue with the manuscript. It drove us to re-investigate relevant literature, and to think carefully about which physical properties likely underpin the radar reflections we observe. We therefore suggest that we amend section 4.1 (lines 116-131) to the following, which we think much improves our explanations for the deep-ice layers, including the reasons for the anisotropy of R1:

"There are several possible explanations for the reflectivity of R1 and R2 including: (i) constructive interference from a series of multiple thin layers (Harrison, 1973; Siegert et al., 1999); (ii) preferred

ice-crystal orientation fabrics (e.g. Matsuoka et al., 2004; Eisen et al., 2007); and (iii) an abrupt spike in the conductivity of the ice column associated with the deposition of volcanic ash (Paren and Robin, 1975; Corr and Vaughan, 2008). These explanations are not mutually exclusive however, and it may be that more than one may act in combination. However, because we observe that the strength of the returned energy from R1 is highly anisotropic, with higher reflectivity in the along-flow orientation (Figures 2, 3, 4, 6 and Supplementary Figures 1-4), we conclude R1 is most likely caused by ice-sheet permittivity rather than conductivity (Fujita et al., 1999; Wang et al., 2018). The depth of R1 rules out ice-density fluctuations, so we therefore attribute the reflection band to crystal orientation fabric. Radar reflection anisotropy associated with crystal orientation fabric has been verified by ice core evidence from Antarctica and Greenland (Eisen et al. 2007, Drews et al. 2012; Li et al. 2018). Deep-ice anisotropic scattering has been observed in convergent ice flow zones, like our study area, in East Antarctica (Matsuoka et al., 2003; Matsuoka et al., 2004). In those studies, anisotropic englacial reflections were attributed to stacked layers of single pole and vertical girdle fabrics observed in the Dome F ice core. Such a model is consistent with our radar observations and ice core observations elsewhere in West Antarctica. A single maximum crystal orientation fabric distribution (i.e. with a fabric characterised by strong vertical c-axes), typical for simple shear would not result in anisotropic scattering, as layer reflectivity would be the same in different survey orientation. A vertical girdle fabric on the other hand is consistent with anisotropic layer reflectivity, as crystals would have an oriented preferred fabric that would likely induce a backscatter response. Evidence for down-ice column evolution of crystal fabric (i.e. from isotropic to anisotropic, and then back to isotropic at depth) is observed in ice cores from West Antarctica (e.g. Gow and Williamson, 1976; Gow and Meese, 2007; Fitzpatrick et al., 2014), with anisotropic crystal fabrics typically associated with ice of last glacial age. However, as stated above, an anistropic crystal fabric (i.e. with a strong single vertical maxima) would not result in an anistropic radar response, so these gradual down-core changes cannot be the explanation for R1. In the Byrd core however, there is evidence for sharply alternating crystal fabrics (i.e. narrow cone to distributed cone and back again) associated with cloudy bands (1-60 mm thick) of glacial-age ice that incorporate tephra (Gow and Williamson, 1976; Horgan et al., 2011). Abrupt alternations in crystal fabric such as these are akin to those proposed as the cause of anisotropic radar scattering in East Antarctica (Fujita et al. 2003; Matsuoka et al., 2003; Matsuoka et al., 2004). Assuming that the cloudy bands in the Byrd core represent the same stratigraphy as R1, then this is a plausible explanation for the radar reflection anisotropy of this layer. The anisotropy cannot be due to directional roughness of layer reflectivity, as the anisotropy is unique to specific layers (Figures 6a and Supplementary Figures 1-4). R2 is also a prominent and strong reflection (Figures 2, 3, and 4), but unlike R1 it is not characterized by anisotropic reflectivity (Figure 6). We consider R2 to represent a layer with a discretely high conductivity, similar to the bulk of internal layers in Antarctica (Siegert, 1999). The anomalously high reflectivity of R2 may represent a pronounced acidity spike, or multiple spikes, in the stratigraphy."

**Additional references for section 4.1:**

Fujita, S., Matsuoka, K., Maeno, H., and Furukawa, T. (2003). Scattering of VHF radio waves from within an ice sheet containing the vertical-girdle-type ice fabric and anisotropic reflection boundaries. Annals of Glaciology, 37, 305-316.

Gow, A.J. and Meese D. (2007). Physical properties, crystalline textures and c-axis fabrics of the Siple Dome (Antarctica) ice core. Journal of Glaciology, 53, 573-584.

Matsuoka, K., T. Furukawa, S. Fujita, H. Maeno, S. Uratsuka, R. Naruse, and O. Watanabe, (2003). Crystal orientation fabrics within the Antarctic ice sheet revealed by a multipolarization plane and dual-frequency radar survey, J. Geophys. Res.,108(B10), 2499, doi:10.1029/2003JB002425.

We suggest that changes made to the conclusions section in response to comments from reviewer 1 have already addressed the issues raised by reviewer 2 regarding "*how anisotropy in the lower layers "modulates ice stream position, structure and dynamics)*".

We anticipate that the proposed changes to section 4.1 will have knock-on impacts to later sections of the paper (e.g. section 4.4 and conclusions section). We will address these issues accordingly if invited by the Editor to prepare a revised version of the manuscript.

*Some smaller points: Lines 27-30: Folds can be generated in an anisotropic material by lateral compression, there is no need for a rheology contrast between two layers. Buckle folding, when a hard layer is embedded in a softer matrix, produces different kinds of folds (parallel folds), and the rheology contrast needed for this is much bigger than the range expected in natural ice. The paragraph reads as if the contrast in rheology is the origin of folding.*
We propose to replace the wording "rheological properties" with "physical properties" here, so that lines 27-30 will read: "Like a structural geology problem, such folds can only be explained by the deformation of ice with contrasting physical properties near the base of the ice sheet. Evidence of variability in physical properties is consistent with ice-penetrating radar observations of a widespread englacial layer characterized by a strongly anisotropic ice crystal fabric, as postulated for ice folds in Greenland (Bons et al., 2016)."

*Lines 103-104: to interpret the fold here as a natural boundary for the stream is a bit circular. The folding is linked to dynamics, so the fold where the dynamic setting is right (shear margin).*
We do not interpret the fold as a natural boundary for the ice stream here. In lines 103-104 we observe the spatial relationship between the fold and the ice plain at the grounding zone of Institute Ice Stream. In fact, in the lower trunk of the Institute, the fold with the hand-shaped reflector is located in the middle of the ice stream trunk, it is not located at the ice stream shear margin (see Figure 5). What we do suggest in lines 103-104 is that given the spatial relationship, the fold *may* influence the location and form of the ice plain in the ice stream grounding zone, which may in turn feedback to the ice stream via buttressing effects. Perhaps the spatial relationship is a coincidence, but we thought it was worth drawing attention to.

*Line 109, Comment on the supplementary figures: It is great to be able to see all the data, but would be really helpful would be a way to quantify the difference between the two profile directions in the relevant depth range and then plot this color-coded on a map.*
We can attempt to do this. A figure of this nature was something that we had considered before submission but decided not to do on technical grounds. Defining the "relevant depth range" in an objective way will likely be problematic, and there are some locations where aircraft elevation was not consistent (see supplement) so there will be some missing data points.

*Lines 112-113: Maybe this is the place to describe what kind of anisotropy you would expect and why.*
As lines 112-113 are part of the results section, we will instead insert our description of the expected anisotropy etc. in section 4.1 of the discussion section, as proposed in response to the general comments of reviewer 2.

*Line 150: Do you mean a band of ice, that is in the lower part the ice should be isotropic again? Or is it a thick basal layer?*
We mean a band of ice (i.e. 'layer' R1). The lack of directionality in the reflection strength of layer R2 suggest that the ice below R1 is isotropic. R1 is the only layer ('band') that displays evidence for anisotropy in the RES data. We therefore propose to reword this sentence to read "Further, given the radar anisotropy observed, the most likely explanation for the folds is that they are caused by a combination of convergent ice flow and the distinct physical (i.e. varying crystal orientation fabric),

and subsequent rheological, properties of the band of ice associated with R1." We suggest removing the references to NEEM community members (2013) and Bon et al., (2016) here, as they are referenced in section 4.4.

*Line 155: As mentioned above, how can you conclude that the fold is influencing the location of the shear margin? In the shear margin there is in general a compressive stress across flow, so it would be the other way around.*
We accept that this is likely, but in response suggest that a feedback mechanism could operate where formation of an englacial fold containing a core with a distinct rheology could reinforce the position of the shear margin. We therefore propose to reword lines 154-157 to: "This spatial correspondence between the fold and shear margin is remarkable and may suggest that the folding of the deep ice modulates the position of the shear margin and controls trunk flow. The fold may therefore play an important role in the ice dynamics of the IIS-MIS catchment."

*Line 165: I don't think that it would be possible to form fractures at the base of an ice sheet. The publication which is cited is about a mountain glacier, only a few hundreds of meters thick. This must be some other form of ductile entrainment of bottom material.*
We propose removing the reference to Woodward and others, 2003, and inserting Winter and others, 2019 https://agupubs.onlinelibrary.wiley.com/doi/full/10.1029/2019GL084012 instead. This paper describes RES evidence for the incorporation of sediment at the base of the West Antarctic Ice Sheet rather than at the base of a mountain/valley glacier.

---

## Author Response (AR1)

**Response to Reviews for: "Large-scale englacial folding and deep-ice stratigraphy within the West Antarctic Ice Sheet" (MS No: tc-2019-245)**

Dear Dr MacGregor,

We are grateful to you and both reviewers for their constructive and helpful reviews of our manuscript. Below we respond (non-highlighted text) to your comments (part1) and those of the reviewers (parts 2&3) (highlighted in grey). Our specific actions in each response are highlighted yellow. We also include a short section (part 4) detailing other significant changes we have made to the manuscript. All references to line numbers refer to the originally submitted manuscript, not the revised manuscript.

**PART 1: Editor Decision: Publish subject to revisions (further review by editor and referees)** (05 Feb 2020) by Joseph MacGregor

**Comments to the Author:**

Thanks for submitting your work to The Cryosphere, and my apologies for the delay in my decision. After review of your MS and the thoughtful referee comments, I consider it likely that the MS will be suitable for publication following revision.

Both referees raise substantive issues with the attribution of the folding to rheology, so I encourage the authors to moderate the text carefully in this regard.

We have responded in detail to the comments of both reviewers in line below. We have made the conclusions less assertive and have moderated the text carefully throughout (e.g. typically referring now to 'physical properties' rather than specifically 'rheology'). In response to a very helpful suggestion from reviewer 2 we have now strengthened aspects of the manuscript that discuss possible explanations for the crystal fabric and anisotropic radar response.

Separately, the supplementary figures are useful, but it is not clear to me – given a finite number of intersections – why the authors do not simply measure the peak reflectivity of the candidate anisotropic layer in both flight directions and then map that quantity. This is important because it is the best opportunity for quantitative assessment of the proposed origin of the layer presently available.

We are currently working on developing this quantitative assessment and deriving a figure from it. This task will likely not be completed in advance of the deadline to submit the corrections however (i.e. today, 4th May), so we thought it wise to submit the manuscript as is with the figure to follow if required. The status of this issue is therefore pending, but we can undertake this work within a couple of days. *This is the only first review item that currently (i.e. at 4th May 2020) remains outstanding*.

In the supplementary figure caption, an important caveat is raised given aircraft altitude, but I find that unconvincing for several reasons:

- A geometric spreading correction for the aircraft altitude difference could easily be done. The difference in Fresnel zone due to altitude difference should be minor, or at least it shouldn't be significant enough to affect the anisotropic behavior of the layer.

- If the authors still find this altitude difference problematic, they could simply not include/interpret/show those flights where that altitude difference is considered significant.

For SI figures where there was a difference in aircraft elevation at the flight crossover location, we have retained the SI plots, but have annotated parts 'a' and 'c' of the figure as 'OFFSET'. We have also added the phrase "...figures with aircraft elevation offsets are clearly marked with 'OFFSET' in parts 'a' and ' c'." to the SI figure captions.

The flights are not always oriented along/across the present-day ice-flow direction.
 We have oriented all the radargrams in the SI figures, and throughout the manuscript, in common directions (i.e. ice flow is left to right for 'down-ice' tie lines, and into page for 'across-ice' flight lines).

**The absolute difference of the IMAFI grid's "+" intersections to ice-flow direction should also be considered as a confounding factor.**

We have acknowledged this issue at the end of the methods section, by inserting the following text:

"Though the survey grid design allows us to undertake a detailed analysis of the pseudo-3D englacial structure of the IIS, the grid layout and orientation of the 2D survey flights was not aligned exactly along and across ice flow. This oblique offset can have implications for the 2D imaging of 3D englacial structures (Bingham et al., 2015). In general, however, our survey grid is approximately aligned along and across flow throughout most of our area of interest. As a result, we refer from this point onwards to 'across-' and 'along-' flow flight lines, although we acknowledge that in many places the flight lines are slightly oblique to both along and across ice flow vectors."

**PART 2: Response to Reviewer 1 for: "Large-scale englacial folding and deep-ice stratigraphy within the West Antarctic Ice Sheet" (MS No: tc-2019-245)**

**Anonymous Referee #1**

**General Comments:**

This paper aims at explaining observations of englacial folding in the lower ice sheet column obtained from radio echo sounding. The deep-ice unit is analysed by mapping anticlines using a high-resolution grid of radio echo sounding data from 2010/11. Analysing the returned radar power at certain cross-overs allows to evaluate if englacial layers show signs of anisotropy. Evaluating different hypothesis explaining the formation of such complex near basal structures such as subglacial freeze-on, varying ice rheology and entrainment of basal material, the authors conclude that the deformation and folding of the near basal unit is related to convergent ice flow and layers of anisotropic ice. The paper shows a nice and extensive data set of radargrams and returned radar power at cross overs. However it is related to the ice flow. The different hypotheses are discussed but I feel that the evidence to choose one hypothesis over an other is lacking in parts. I'm not sure what the goal of the paper is - presenting the large-scale englacial folding or resolving how such a structure is obtained. I don't think the latter point is achieved.

We thank reviewer 1 for their review and helpful suggestions to improve the manuscript. We very much appreciate the description of our "nice and extensive dataset of radargrams". We believe that this dataset is unique for a modern SAR-processed radio-echo sounding survey in Antarctica. There are certainly few airborne RES surveys of an appropriate orientation and such closely spaced survey lines across the onset zone of a major Antarctic ice stream system like the Institute-Möller ice stream complex.

We would disagree with the comment that it is "...rather difficult to orient oneself between the different figures to really understand how the data is related to the ice flow...", as we have provided maps of ice flow in four of the six figures, whilst the location of figure 3d (figure 3 is one of the two figures that does not have its own ice flow map) is shown on figure 1c (formerly 1d), where ice flow contours are shown above a map of bed elevation. However, we were happy to adopt many of the later specific suggestions from reviewer 1 for our figures (i.e. labelling tie and across lines, adding the

location of radargrams in figure 2 to figure 1a, and having additional area boxes in figure 1). We hope that accepting these should also address reviewer 1's general comment.

Reviewer 1 states that they are "not sure what the goal of the paper is - presenting the large-scale englacial folding or resolving how such a structure is obtained". The primary purpose of the manuscript is to report the discovery of the large-scale englacial folding AND the unusual deep ice stratigraphy (i.e. extensive thick high reflectivity layers at depth) across the study area. Secondary goals of the manuscript are to infer the physical properties of these folds and layers; to develop hypotheses for how the layers, and associated structures, might form; and to assess any potential glaciological impact. Whilst our "nice and extensive dataset" can be used to identify the englacial structures, and, in combination with ice velocity datasets, can be used to describe its spatial relationship with ice flow, we acknowledge that it does have its limitations. Whilst we can infer some physical properties (i.e. strong preferred crystal fabric) from the data, we lack direct measurements of physical properties (e.g. from ice cores corresponding to the airborne RES survey) to 'calibrate' the radar data, so we are unable, at present, to resolve "how such a structure is obtained". However, from the available datasets, and with reference to glaciological theory, combined with existing observations elsewhere in Antarctica and Greenland, we have been able to develop initial explanations for "resolving how such a structure is obtained". Full determination of the processes responsible for the englacial structures is beyond the current dataset and therefore this paper, however, and will need to await future direct investigations and observations (e.g. coring and geophysical measurements) designed specifically for addressing this question. We note that the current RES dataset was not designed, or acquired, to answer such a question; it was designed to test for the presence or absence of subglacial sediments beneath Institute Ice Stream. The discovery of the englacial structures and layers was therefore serendipitous. Given this context, we would argue that we have successfully reported the englacial layering and structures, inferred possible physical properties, proposed possible processes responsible for fold formation, and assessed the glaciological implications of the folding. That we have not fully resolved "how such a structure is obtained" is not, in our opinion, a major issue. We have undertaken the best assessment and interpretation of the evidence possible, given the data currently at our disposal.

**Specific Comments:**

**- Line 4: this process might also involve freeze-on of basal water?**

We acknowledge that local freeze-on of basal water is possible, but the widespread and extensive nature of the deep-ice facies we report is inconsistent with the freeze-on of basal water being an important process in its formation. The sentence in line 4 reports that we observe little evidence for basal water within our study area (e.g. there are few definite subglacial lakes) as detailed later in the manuscript. We have therefore made no change here. Additional discussion of this issue is outlined in our comments on lines 134-140 below.

- Line 18: the citation of Dow et al. 2018 is not entirely correct in this context - as the impact of freezeon units on ice-sheet flow and dynamics has not been investigated by that paper as they use a subglacial hydrology model and not an ice-flow model.

We have simply removed the reference to this paper at this point in the manuscript (though we retain it elsewhere).

- Line 27-28: This sentence 'Like a structural geology problem, ...' is very assertive statement and I wonder why you are so sure about this. Does this come from the literature (but then references are missing) or does it come from your own findings (then say so)? Here in the introduction it seems to be at the wrong place.

We have inserted references to papers by MacGregor and others, JGR, 2015 https://agupubs.onlinelibrary.wiley.com/doi/10.1002/2014JF003215 and Hudleston, Journal of

Structural Geology, 2015 https://www.sciencedirect.com/science/article/pii/S0191814115300365 here. Both these papers draw attention to similarities between ice sheet englacial structures and structural geology, and we hope that their inclusion will address reviewer 1's comment.

- Line 57: You mention the ice thickness in this sentence but not how thick the lower-ice column is. It is mentioned in the abstract but here would be a good place too.

The thickness of the lower ice column varies significantly (i.e. 100-1000 m). We have inserted the following to the end of the sentence "...up to ~1000 m thick".

- *Line 59: I think not only Figure 3 but also Figure 6 could be referred to.* We agree. We have added the reference to figure 6 here.

- Line 60-61: R2 is referred to as a band of prominent reflection that "sometimes diverges and bifurcates, becoming a series of 3-4 layers". Is it the band that diverges and bifurcates or rather the series of layers within? I suggest a clearer wording.

We have reworded lines 59-61 to: "Beneath R1, within the low-reflectivity lower-ice column, a second highly reflective englacial reflection (R2) is observed. R2 sometimes diverges and bifurcates, becoming a series of 3-4 layers (Figure 2b)."

- Line 64: I'm not sure if instead of Figure 2a you rather mean Figure 1d? It is not clear where exactly you refer to in Figure 2a - which tributary and wich lateral boundary (I guess in the center - parallel to K'K and J'J)?

We have referred to figure 1c (formerly 1d) instead of 2a.

- Line 67: Can you give examples where the amplitudes of up to 40% are seen? Maybe highlight in one of the Figures so that it is clear to what you refer to.

We have shifted the annotations of the englacial folds so that they mark the peak of the fold axis in the ice column. This now shows where the amplitudes of the folds account for ~40% of the ice column on figures 5c and 5d.

- Line 69: For the statement that rather bed conformable undulations than folds are found along flow Figure 2b-c might not be the best example. Comparing the transects where the anticlines are marked (Figure 1d and 3a-b) it is obvious that the chosen along profile in 2 is not crossing many marked anticlines. Along flow one would expect not to see much of a fold that is oriented along flow (e.g. as shown in Figure 4 it is difficult to see the along flow pattern).

The purpose of referring to 2b-c here is that these radargrams demonstrate that there are no folds or overturnings in radar data acquired in that orientation. They are presented as typical examples of the survey data and show that the deep reflective layers are very extensive (i.e. over distances of ~250 km or more) and bed conformable. If referring to figures 2b-c in line 69 remains an issue we can simply remove reference to 2b-c, as 3c demonstrates the same point just not over such a large spatial area. However, we would prefer to leave line 69 as is, and have therefore made no change. The evidence for bed conformable layering down flow is inconsistent with localised basal freeze-on being behind the formation of the deep ice units. Such a process could not explain this layer stratigraphy and structure.

- Line 93-94: This surface flow stripe is not visible in any of the figures (neither main text nor supplement). It would be nice to see the mapped fold together with the flow stripe.

The position of the surface flow stripe was mapped in the original figure 5a, but we have added an additional subplot (comparable to the size and area of the original 5a, now 5b) to figure 5 that shows RADARSAT surface imagery of the ice stream trunk with the flow stripe and areas of surface crevassing visible. This should provide the 'raw data' of the surface flow stripe requested by reviewer 1.

- Line 97: Could you highlight where the "significant shift in basal reflectivity" is seen? This would make it much clearer for the reader what you mean.

We have added dashed red lines to the radargrams in figure 5 to highlight where for those individual radargrams the bed is particularly bright (i.e. there is high basal reflectivity).

- Line 98: Same as above - highlight the onset zone of IIS so that it is very clear to what you are referring to.

We have amended the sentence to: "...that extends from the onset zone where ice is typically flowing at 50-125 m yr-1, into the trunk...". This should provide clarity without overly cluttering figure 5.

-Lines 102-103: From the text/figures it is not clear to what exactly you are referring to. E.g.: - where is "grid-SW margin" to be found? - where is the ice plain? Mark/label it so that it is clear. I can guess but rather would like to be sure. Add these explanations to the overview Figure 1.

The ice plain is located just beyond the grid-north limit of the original figure 5a. We have expanded the area of figure 5a to include it. We have annotated the ice plain on the figure. From this annotation and the map, it should be clear where the grid-SW margin of the ice plain is.

- Line 113: I'm not sure that the observation of no high relief in basal topography is enough to say that anisotropy is not forming in soft layers due to enhanced stresses on the stoss-face of basal hills. One can imagine that if locally basal freeze-on exists, then the plume of accreted ice pushing into the ice columne might act as an obstacle to the meteoric ice flow.

We have simply deleted the sentence at the end of the paragraph (lines 112-113) that dismisses this idea (i.e. the sentence "We do not observe any high relief basal topography in our study region, hence this specific explanation is unlikely to hold here." has been deleted).

- Lines 133-134: The usage of deep-ice units in this sentence is not clear as it seems to be used in two different contexts: 1) as the basal freeze-on units of accreted ice from the bed raising into the ice sheet as hypothesised by Bell et al, 2011; and 2) as the unit of ice between R1 and the bed. So I wonder what in this paper is understood by the "deep-ice units". It is defined as the ice between R1 and the bed. I don't think you expect the whole unit to be frozen on by basal water. With basal freeze-on one would expect the to have some local areas where "material" is rising into the ice sheet and advecting downstream - shaping the meteoric layers above.

We confirm that for our study area, the "deep-ice unit" is the body of ice between R1 and the bed. To limit confusion in lines 133-134, we have inserted "basal ice" instead of "deep-ice unit" when referring to the paper on basal ice features in East Antarctica by Bell et al. 2011. The sentence now reads: "The 'freeze-on' hypothesis for the formation of basal ice (Bell et al., 2011) cannot explain the deep-ice unit and incorporated layers (i.e. R1 to the ice sheet bed) for four reasons:......".

- Lines 134-140: The arguments against basal freeze-on, of slow-flowing and thin ice likely frozen to the bed, as well as fast-flowing ice over a wet bed, do not entirely exclude the formation of basal freezeon plumes. In order to exclude basal freeze-on the possibility of water needs to be excluded. Where is the evidence that there is no water at the base of the ice-sheet? Is it visible in the radar data? It would make sense to map the anticlines together with the main topographically-constrained subglacial drainage network to evaluate if basal freeze-on is possible. Further the ice velocity itself cannot exclude the formation/existence of substantial freeze-on units.

We suggest that including an additional figure showing the relationship of the mapped anticlines with the subglacial hydrological drainage network in the manuscript is unnecessary. Such a figure could not explain or dismiss the widespread existence of the strongly reflective deep ice layers R1 and R2 across the study area (see lines 52-55 of the manuscript), even in areas that are almost certainly underlain by cold based ice. Furthermore, GIS-derived subglacial hydrological models assume a fully warm-based

ice sheet, which is unrealistic. Maps of the subglacial hydrological pathways of the Institute Ice Stream have previously been published in Jeofry et al., Earth System Science Data, 2018a <a href="https://doi.org/10.5194/essd-10-711-2018">https://doi.org/10.5194/essd-10-711-2018</a> (Figure 3); Jeofry et al., Nature Communications, 2018b <a href="https://doi.org/10.1038/s41467-018-06679-z">https://doi.org/10.1038/s41467-018-06679-z</a> (Supplementary Figure 1); and Le Brocq et al., Nature Geoscience, 2013 <a href="https://doi.org/10.1038/ngeo1977">https://doi.org/10.1038/ngeo1977</a> (Supplementary Figure 511). All show subglacial water routing obliquely through our focused study area, and obliquely to the mapped folds. In addition, there is relatively limited evidence for subglacial water in RES data across the catchment; there are very few subglacial lakes for example. In lines 139-146 of the original manuscript submission, we describe the limited evidence (in our, and others', RES data) for "significant ponding of subglacial water" in the study area. We have therefore taken no action in response to this suggestion from reviewer 1, except to insert references to the work of Jeofry et al., 2018a and 2018b in the sentence "(3) IIS has a well-defined and efficient topographically-constrained subglacial drainage network, without widespread stores of subglacial water to act as water sources (Wright and Siegert, 2012; Jeofry et al., 2018a, 2018b)".

With regards to the questions "Where is the evidence that there is no water at the base of the icesheet?" and "Is it visible in the radar data?" we refer the reviewer to lines 139-146 of the manuscript, where we describe the limited evidence (in our, and other, RES data) for "significant ponding of subglacial water" in the study area.

- Lines 147-149: I don't understand the argument that because R1 and R2 have a consistent stratigraphic position "the structures and extent of the deep-ice structures must be the result of the deformation and localized folding of meteoric ice". How is this meant - a stratigraphic position in the vertical ice column? They will always be on top of each other - unless one is melted away. E.g. basal freeze-on would lead to the same result just pusing up the entire ice column - with the vertical distance between the layers diminishing.

It is true that basal freeze-on would lead to the same result by pushing up the entire ice column, but it is physically unrealistic for this to have occurred over such a widespread area as we observe R1 and R2. The volumes of water required would be, quite simply, enormous and freeze-on rates would have to be consistent right across the study area. We might expect freeze-on locally (and we do not dispute the possibility of localised basal freeze-on), but not over the extensive area that we map the layers (see Lines 52-55 and Figure 1). In addition, freeze-on cannot be responsible for the formation of the basal ice layers over high subglacial mountain ranges, e.g. the Ellsworth Subglacial Highlands, where the ice is cold based and has likely been for millions of years. Therefore, the only physically realistic explanation for the consistent stratigraphic position of R1 and R2 in the ice column, over such a vast geographical area, is that they reflect layers that were initiated at the surface by meteoric deposition (i.e. of snow and/or volcanic ash), but that will have undergone physical modification (i.e. to crystal fabric orientation) in the ice column during burial and strain. We have referenced figure 1, in addition to figure 2, in line 147 however, to direct the reader to the extensive geographical distribution of the deep ice layers.

- Line 148: What exactly is your deep ice structure? In parts it sounds that it is all the ice below R1? Or do you mean the layer pattern at depth? Is there a difference between units and structure? By deep ice we mean all the ice between R1 and the bed. Units are the layers of ice (like geological units), the deep ice structure refers to the geometry of the units (e.g. folds) in the deep ice.

- Lines 149-151: It is not clear to me how the anisotropy observed in a layer leads to the folding of the ice unit below. In the paper by Bons et al., 2016 it is not one anisotropic layer that leads to folding but a body of anisotropic ice.

In response to a comment from reviewer 2, we have amended lines 149-151 to: "Further, given the radar anisotropy observed, the most likely explanation for the folds is that they are caused by a

combination of convergent ice flow and the distinct physical (i.e. crystal orientation fabric), and consequent rheological, properties of the band of ice associated with R1.". As is clear from figure 6 (and multiple other examples in the supplementary information) R1 is a broad band (described in line 58 as "up to 200 m" thick) that is rather different from the narrow reflections we typically think of as englacial layers observed in RES data. As such, R1 is more a body or package of ice than a single layer (and therefore potentially like Bons et al, 2016). Furthermore, we do not know the properties of most of the ice below R1 as, unlike in Greenland, as we do not have direct observations (e.g. from ice cores) to calibrate against. We can infer from the RES data that R1 is anisotropic, and that R2 is likely isotropic. However, we do not know the physical properties of the rest of the deep ice units as it is typically an echo-free zone below R1. It may be anisotropic, either in places or throughout, but we cannot determine that from the currently available data.

 Lines 169-171: Why would the spiraling flow of basal ice only happen at that specific location and why only on one side of the ice stream onset?
 We have deleted lines 169-171 and the reference to Schoof and Clarke 2008.

- Lines 173-176: "strongly contrasting physical and rheological properties of glacial and interglacial ice" might not be the only explanation that basal units are" folded, sheared and overturned". We accept that englacial layers in Greenland and Antarctica can be deformed by the freeze-on of basal ice (which we assume reviewer 1 is alluding to here), but based on the currently available literature.

ice (which we assume reviewer 1 is alluding to here), but based on the currently available literature for Greenland (which we cite in lines 175-176) the most widespread accepted explanation for such folding is the strong contrast in physical and rheological properties between glacial ('soft') and interglacial ('hard') ice. We have therefore made no change to this sentence.

- Line 203: I do not agree that you "demonstrated" that the deep-ice units have different physical properties than the ice above. You showed that one band does show in some places anisotropic behaviour.

We have reworded this sentence to: "We have demonstrated the presence of an extensive package of deep-ice units beneath the Institute and Moller ice streams. At least one layer in the deep ice has physical properties (i.e. ice crystal orientation fabric, rheology) significantly different to the upper ice column."

- Line 204-206: I'm not happy with the statement in this sentence. The paper does not show that convergent flow heavily deforms deep-ice units, that this process leads to the formation of large-scale englacial folds and that these folds modulate ice-stream position structure and dynamics.

We have reworded this sentence to: "At the lateral boundary of the onset of enhanced ice flow of IIS, where ice flow is convergent, these deep-ice units have been heavily deformed. Deformation has led to the development of large-scale englacial folds that may modulate ice-stream position, structure and dynamics."

- Conclusions: The conclusion seems to me in parts too assertive. In my view the study is not really conclusive how this structures are really formed. I agree not so much with the top part but agree with the bottom half. It is still not clear what exactly is meant by deep-ice unit.

We have now addressed this issue with the changes made to lines 203, and lines 204-206 (see above). Those statements are now less "assertive" than they were. We acknowledge that our study may not be conclusive as to how the structures formed, but that reflects the serendipitous nature of the discovery and the datasets currently available. More conclusive understanding will require additional work (i.e. new geophysical/ice core observations and/or numerical modelling) beyond this 'discovery' paper. As stated previously, the deep ice unit is all the ice between R1 and the bed. We have made no additional changes to the conclusions section in response to this comment by reviewer 1.

- Figures 1: The figures in the paper are not very well related to each other. It is difficult to see where the transects are taken relative to ice velocity, bed topography surface slope and the mapped anticlines. I suggest that the boxed area of 1d is not only shown in 1a but everywhere in 1. Ideally 1d would show bedtopography upstream of the section in 1d. Help reader to orientate by defining the orientation of the grid used in the figure.

We have added the boxed area shown in 1d (now 1c) to 1a & 1b. We have also expanded the extent of 1d (now 1c) up-ice. However, we are unclear what the reviewer means by *"Help reader to orientate* by defining the orientation of the grid used in the figure.". The orientation of 1a-c should be clear from the inset figure in 1a. We would be happy to modify the figure in response to additional clarification on this, however.

- Figure 2: Where is this line (2a) in Figure 1 - maybe useful to show the tie line 1 and the across line 1 (or other number) as to allow reader to orient oneself in the overview figure. An overview of the shown radar profiles would be nice (see in Figure 1).

We have added the location of the radar data shown in 2b-c to figure 1a as well as showing it in 2a. We have inserted this sentence to the figure 1 caption "White line is survey tie line 9, shown in figure 2."

- Figure 3: What is the criteria for a fold - in 3b I can't see a fold in the radargram where the third "A" from left (from Y1 towards Y2) is. Further comparing the mapped "A" with Figure 1d it seems to me that sth. is wrong. In 1d "HSR" is the second from right and not the third as shown in radargram and the spacing between the folds is different than in overview Figure 1d.

Folds were mapped where the englacial layers were folded and were not bed conformable. In the case of 3b, we were somewhat overenthusiastic in our labelling of the third A from the left, based on our down-ice mapping of a fold in parallel radar lines (i.e. we tracked it back (up-ice) to that location on the 3b radargram from down-ice radargrams). We have deleted the third "A" from the left.

With regards to the 'HSR', we made an error in the manual labelling of this feature in figure 3b. Further up-ice, where 3b is located, it does become harder to confidently identify the folds. However, through identification of the diagnostic form of a package of whirlwinds in the surrounding ice column, we were able to correctly identify the HSR at 36 km along the radargram (see Figure 3b). We have now moved the labelling of the HSR from the 3rd to the 1st fold from the right.

 Figure 4: Numbering the radar lines would be helpful (eventhough one can deduce the numbering) especially in context with Figure 1 (if there is some numbering).
 We have added numbering and labelling of survey lines to Figure 4 and 1a.

- Figure 5: Distance taken from where for Figures 5b and c? Why not just show the "length" of these two radargrams (as a distance is o.k.). Why is the "signature of englacial folding" (caption line 5) not mapped/marked? What is the criteria for mapping these folds?

The distance is from the start of the radargram. This was an error, which we have now corrected, so that the radargrams x-axes are from 0-25 km. We have added additional arrows to highlight the two most obvious englacial folds apparent in figure 5. Mapping of the folds is based on obvious deviation of englacial layering from the bed profile, and identification of folding in up-ice radargrams.

- Figure 6: in (b) why not mark in the white circle the along line in blue and across in red. In (c) and (d) it's not entirely clear if the colour of the vertical line represents the colour of the crossing transect. Maybe it makes more sense to mark in (c) in red where the across transect crosses and vice versa in (d). The caption explaining this is slightly confusing. For (c) and (d) it's not clear where -5 and 5 is e.g. is (c) going downstream from -5 to 5 or upstream? Are the across profiles (d) always oriented the same way? An orientation would help.

Because of the scale of the figure we were concerned that two 10 km-long plots of the radargram locations would be lost, hence why we originally opted for the circle and cross to demonstrate location. We have updated the figure replacing the circle/cross with a white circle overlain by the location and extent of the radargrams that are shown in 6c and 6d, as suggested by the reviewer.

We have swopped the colours of the vertical lines in the radargrams. This makes sense.

The radargrams in c and d are oriented along the direction of the survey flight. Fortunately, in this case the survey flights were in directions that are logical for the display of the radargrams in this figure. Therefore, we have amended the figure caption to state that ice flow is left to right for 6c (i.e. radargram is from bottom to top of survey grid), and into the page for 6d (i.e. radargram is left to right across survey grid). We have oriented all the radargrams in the supplementary figure section to match these orientations.

- Figures in Supplement: I'm not sure if the across profiles all (d)'s are always along the same orientation. Especially when looking at "HSR" in along tie10 it seems to me that it "jumps" from one across profile to the other. It would be nice to have the same geographical orientation for all along and across profiles at all times. As mentioned above it would be nice to know where the tie lines are. No, the across profiles were not always in the same orientation, as they were oriented in the direction of the survey flight, and different survey flights were flown in different directions. We have now oriented all the radargrams in the same way, i.e. ice flow from left-to-right for down flow TIE lines (radargram oriented grid south to grid north), and ice flow into page for across flow lines (radargram oriented grid west to grid east).

Tie and across lines have now been labelled on figure 1 and 4 of the manuscript (see response to comment by reviewer 1 on figure 4).

Technical corrections: - Caption 2: Line 2 - "ubiquity and widespread" is this not the same message? We have deleted "ubiquity"

Caption 2: Line 2 - band of ice-layers rather than "deep-ice layers"
 We have made this change.

- Caption 2: Line 4 - "thin black line" of grounding line is not easy to differentiate from the "thin black lines" of the survey grid.

Agreed, we have changed the grounding line to a white line, and the white radar line to a thick black line.

- *Caption 2: Line 5* - "bifurcation of R2 into 3-4 layers" happens also earlier around 50 km. Agreed. We had not spotted this. We have amended the caption accordingly.

- Caption 2: Line 5 - in (c) mention tie line 9 as used in Figure 4 and supplementary figures. Agreed. We have inserted the following sentence to the caption of figure 6: "Radargrams in 2b and 2c are radar survey 'tieline' 9, which is also shown in Figure 4 and Supplementary Figure 1."

- *Caption 2: Line 6 - replace "show" with "shown".* Agreed. We have amended the caption accordingly.

- Caption 5: Line 4 - the transects are not "perpendicular" to the ice flow rather at an angle of 45 degrees oblique to ice flow.

Agreed. We have amended the caption accordingly.

**PART 3: Response to Reviewer 2 for: "Large-scale englacial folding and deep-ice stratigraphy within the West Antarctic Ice Sheet" (MS No: tc-2019-245)**

**Anonymous Referee #2**

This paper presents an extensive data set of airborne radar across the tributaries of the Institute Ice Stream. The high-quality data makes it possible to track folds in the lower part of the ice column and investigate the nature of a particular reflector band with a directional dependency of reflection strength, indicating crystal anisotropy in the lower layers of ice. Different ice rheology due to anisotropy and the redistribution of this ice in the folding process is identified as playing a role for the organization of inflow to the ice stream. This is an interesting data set, with most profiles published in the supplements, and highlights the importance of considering ice rheology and anisotropy when trying to understand large scale ice flow.

**Technically the manuscript is sound and well written. But I would like to raise some points which could be improved in the final version.**

We thank reviewer 2 for their review and helpful suggestions to improve the manuscript. We very much appreciate the recognition that our data is "high-quality", "interesting" and "highlights the importance of considering ice rheology and anisotropy when trying to understand large scale ice flow". We are also delighted that they find that "Technically the manuscript is sound and well written".

Technically the manuscript is sound and well written. But I would like to raise some points which could be improved in the final version. My main concern is how the anisotropy of the ice crystal fabric is discussed. It is known from ice core data, seismic studies and also from numerical modelling that anisotropic ice is to be expected in ice sheets, and that this is linked to the dynamic setting and the deformation the ice has been subject to, as well as the influence of impurities in the ice. This is briefly mentioned here. However, there are different types of anisotropy, which are again linked to certain deformation regimes. To me it is not clear from this manuscript what kind of change in crystal fabrics is causing the reflection package described. The directional dependency of the reflector strength would indicate that it is a girdle type, typical for extensional flow, as a single maximum distribution (typical for shear) is symmetric to the vertical and therefore would not be different in profiles at different angles. I think it would be essential to discuss the different types of anisotropy and how they are linked to dynamics, to be able to interpret the influence on ice stream flow of the anisotropic ice package.

As it stands now I would not agree that the data support the strong conclusion that is has been shown how anisotropy in the lower layers "modulates ice stream position, structure and dynamics). I do not disagree in general, I just think the authors have to be more convincing in their line of arguments. In summary I think there is need for a better interpretation of the links between anisotropy, ice dynamic setting, and folding structures.

We are very grateful to reviewer 2 for highlighting this issue with the manuscript. It drove us to reinvestigate relevant literature, and to think carefully about which physical properties likely underpin the radar reflections we observe. We have amended section 4.1 (lines 116-131) to the following, which we think much improves our explanations for the deep-ice layers, including the reasons for the anisotropy of R1:

"There are several possible explanations for the reflectivity of R1 and R2 including: (i) constructive interference from a series of multiple thin layers (Harrison, 1973; Siegert et al., 1999); (ii) preferred ice-crystal orientation fabrics (corresponding to power anisotropy for a specific radar layer) (e.g.

Matsuoka et al., 2004; Eisen et al., 2007); (iii) Birefringent propagation (corresponding to power oscillations as a function of ice depth) (Fujita et al., 2006); and (iv) an abrupt spike in the conductivity of the ice column associated with the deposition of volcanic ash (Paren and Robin, 1975; Corr and Vaughan, 2008). These explanations are not mutually exclusive however, and it may be that more than one may act in combination. However, because we observe that the strength of the returned energy from R1 is highly anisotropic, with higher reflectivity in the along-flow orientation (Figures 2, 3, 4, 6 and Supplementary Figures 1-4), we conclude R1 is most likely caused by (ii) or (iii) (Fujita et al., 1999; Wang et al., 2018). The depth of R1 certainly rules out ice-density fluctuations. In general, it is difficult to separate-out (ii) and (iii). However, for the case of R1, we can assume that anisotropic scattering is the dominant cause of power anisotropy is localized for a set of radar layers). We therefore attribute R1 to a crystal orientation fabric.

Radar reflection anisotropy associated with crystal orientation fabric has been verified by ice core evidence from Antarctica and Greenland (Eisen et al. 2007, Drews et al. 2012; Li et al. 2018; Montagnat et al., 2014). Deep-ice anisotropic scattering has been observed in convergent ice flow zones, like our study area, in East Antarctica (Matsuoka et al., 2003; Matsuoka et al., 2004). In those studies, anisotropic englacial reflections were attributed to stacked layers of single pole and vertical girdle fabrics observed in the Dome F ice core. Such a model is consistent with our radar observations and ice core observations elsewhere in West Antarctica. A single maximum crystal orientation fabric distribution (i.e. with a fabric characterised by strong vertical c-axes), typical for simple shear would not result in anisotropic scattering, as layer reflectivity would be the same in different survey orientation. A vertical girdle fabric on the other hand is consistent with anisotropic layer reflectivity, as crystals would have an oriented preferred fabric that would likely induce a backscatter response. Evidence for down-ice column evolution of crystal fabric (i.e. from isotropic to anisotropic, and then back to isotropic at depth) is observed in ice cores from West Antarctica (e.g. Gow and Williamson, 1976; Gow and Meese, 2007; Fitzpatrick et al., 2014), with anisotropic crystal fabrics typically associated with ice of last glacial age. However, as stated above, an anistropic crystal fabric (i.e. with a strong single vertical maxima) would not result in an anistropic radar response, so these gradual down-core changes cannot be the explanation for R1. In the Byrd core however, there is evidence for sharply alternating crystal fabrics (i.e. narrow cone to distributed cone and back again) associated with cloudy bands (1-60 mm thick) of glacial-age ice that incorporate tephra (Gow and Williamson, 1976; Horgan et al., 2011). Abrupt alternations in crystal fabric such as these are akin to those proposed as the cause of anisotropic radar scattering in East Antarctica (Fujita et al. 2003; Matsuoka et al., 2003; Matsuoka et al., 2004). Assuming that the cloudy bands in the Byrd core represent the same stratigraphy as R1, then this is a plausible explanation for the radar reflection anisotropy of this layer. The anisotropy cannot be due to directional roughness of layer reflectivity, as the anisotropy is unique to specific layers (Figures 6a and Supplementary Figures 1-4). R2 is also a prominent and strong reflection (Figures 2, 3, and 4), but unlike R1 it is not characterized by anisotropic reflectivity (Figure 6). We consider R2 to represent a layer with a discretely high conductivity, similar to the bulk of internal layers in Antarctica (Siegert, 1999). The anomalously high reflectivity of R2 may represent a pronounced acidity spike, or multiple spikes, in the stratigraphy."

**Additional references for section 4.1 that have been added to reference list:**

Fujita, S., Matsuoka, K., Maeno, H., and Furukawa, T. (2003). Scattering of VHF radio waves from within an ice sheet containing the vertical-girdle-type ice fabric and anisotropic reflection boundaries. Annals of Glaciology, 37, 305-316.

Fujita, S., Maeno, H., and Matsuoka, K.: Radio-wave depolarization and scattering within ice sheets: a matrix-based model to link radar and ice-core measurements and its application, Journal of Glaciology, 52, 407-424, https://doi.org/10.3189/172756506781828548, 2006.

Gow, A.J. and Meese D. (2007). Physical properties, crystalline textures and c-axis fabrics of the Siple Dome (Antarctica) ice core. Journal of Glaciology, 53, 573-584.

Matsuoka, K., T. Furukawa, S. Fujita, H. Maeno, S. Uratsuka, R. Naruse, and O. Watanabe, (2003). Crystal orientation fabrics within the Antarctic ice sheet revealed by a multipolarization plane and dual-frequency radar survey, J. Geophys. Res., 108(B10), 2499, doi:10.1029/2003JB002425.

Montagnat, M., Azuma, N., Dahl-Jensen, D., Eichler, J., Fujita, S., Gillet-Chaulet, F., Kipfstuhl, S., Samyn, D., Svensson, A., and Weikusat, I.: Fabric along the NEEM ice core, Greenland, and its comparison with GRIP and NGRIP ice cores, The Cryosphere, 8, 1129–1138, https://doi.org/10.5194/tc-8-1129-2014, 2014.

We suggest that changes made to the conclusions section in response to comments from reviewer 1 (lines 203, lines 204-206) have already addressed the issues raised by reviewer 2 regarding "how anisotropy in the lower layers "modulates ice stream position, structure and dynamics)". No further changes have therefore been made to this comment.

Some smaller points: Lines 27-30: Folds can be generated in an anisotropic material by lateral compression, there is no need for a rheology contrast between two layers. Buckle folding, when a hard layer is embedded in a softer matrix, produces different kinds of folds (parallel folds), and the rheology contrast needed for this is much bigger than the range expected in natural ice. The paragraph reads as if the contrast in rheology is the origin of folding.

We have replaced the wording "rheological properties" with "physical properties" here, so that lines 27-30 now read: "Like a structural geology problem, such folds can only be explained by the deformation of ice with contrasting physical properties near the base of the ice sheet. Evidence of variability in physical properties is consistent with ice-penetrating radar observations of a widespread englacial layer characterized by a strongly anisotropic ice crystal fabric, as postulated for ice folds in Greenland (Bons et al., 2016)." References by MacGregor & Hudleston were inserted into lines 27-30 in response to a comment from reviewer 1.

Lines 103-104: to interpret the fold here as a natural boundary for the stream is a bit circular. The folding is linked to dynamics, so the fold where the dynamic setting is right (shear margin).

We do not interpret the fold as a natural boundary for the ice stream here. In lines 103-104 we observe the spatial relationship between the fold and the ice plain at the grounding zone of Institute Ice Stream. In fact, in the lower trunk of the Institute, the fold with the hand-shaped reflector is located in the middle of the ice stream trunk, it is not located at the ice stream shear margin (see Figure 5). What we do suggest in lines 103-104 is that given the spatial relationship, the fold \*may\* influence the location and form of the ice plain in the ice stream grounding zone, which may in turn feedback to the ice stream via buttressing effects. Perhaps the spatial relationship is a coincidence, but we believe that it is worth drawing attention to. No change made.

Line 109, Comment on the supplementary figures: It is great to be able to see all the data, but would be really helpful would be a way to quantify the difference between the two profile directions in the relevant depth range and then plot this color-coded on a map.

We are currently working on developing this quantitative assessment and deriving a figure from it. This task will likely not be completed in advance of the deadline to submit the corrections however (i.e. today, 4th May), so we thought it wise to submit the manuscript as is with the figure to follow if required. The status of this issue is therefore pending, but we can undertake this work within a couple of days. *This is the only first review item that currently (i.e. at 4th May 2020) remains outstanding*.

*Lines 112-113: Maybe this is the place to describe what kind of anisotropy you would expect and why.*

As lines 112-113 are part of the results section, we will instead insert our description of the expected anisotropy etc. in section 4.1 of the discussion section, as proposed in response to the general comments of reviewer 2. Please see our response to the 'general comments' of reviewer 2 above. Extensive changes have been made to section 4.1 that describe the likely nature of the anisotropy in response to general comments made by reviewer 2. We therefore make no further change.

**Line 150: Do you mean a band of ice, that is in the lower part the ice should be isotropic again? Or is it a thick basal layer?**

We mean a band of ice (i.e. 'layer' R1). The lack of directionality in the reflection strength of layer R2 suggest that the ice below R1 is isotropic. R1 is the only layer ('band') that displays evidence for anisotropy in the RES data. We have reworded this sentence to read "Further, given the radar anisotropy observed, the most likely explanation for the folds is that they are caused by a combination of convergent ice flow and the distinct physical (i.e. varying crystal orientation fabric), and subsequent rheological, properties of the band of ice associated with R1." We have removed the references to NEEM community members (2013) and Bon et al., (2016) here. They are still referenced in section 4.4.

Line 155: As mentioned above, how can you conclude that the fold is influencing the location of the shear margin? In the shear margin there is in general a compressive stress across flow, so it would be the other way around.

We accept that this is likely, but in response suggest that a feedback mechanism could operate where formation of an englacial fold containing a core with a distinct rheology could reinforce the position of the shear margin. We have therefore reworded lines 154-157 to: "This spatial correspondence between the fold and shear margin is remarkable and may suggest that the folding of the deep ice modulates the position of the shear margin and controls trunk flow. The fold may therefore play an important role in the ice dynamics of the IIS-MIS catchment."

Line 165: I don't think that it would be possible to form fractures at the base of an ice sheet. The publication which is cited is about a mountain glacier, only a few hundreds of meters thick. This must be some other form of ductile entrainment of bottom material.

We have removed the reference to Woodward and others, 2003, and have inserted Winter and others, 2019 https://agupubs.onlinelibrary.wiley.com/doi/full/10.1029/2019GL084012 instead. This paper describes RES evidence for the incorporation of sediment at the base of the West Antarctic Ice Sheet rather than at the base of a mountain/valley glacier.

**PART 4: Additional significant changes made by the authors**

- We felt that it was important to clarify in the methods section which 'flavour' of PASIN radar data we were using. Therefore, we have inserted the following sentences: "The data used were from the chirp mode of the PASIN system (Jeofry et al., 2018a; Ashmore et al., 2020). We did not use PASIN pulse channel data in this study."
- 2. We have updated section 4.4 to account for the publication of Ashmore et al. 2020, which strengthens the dating of the deep ice units. The central part of this section now reads: "A new radiostratigraphy across West Antarctica (Ashmore et al., 2020) enables us to infer the likely age of the deep ice units in the IIS-MIS. The best 'direct' constraint on the age of the deep-ice package is from the dated radar englacial layer tied to the Byrd ice core (Siegert and Payne, 2004; Ashmore et al., 2020). Linking the radar stratigraphy of the IIS-MIS catchment to the radar transect of Siegert and Payne (2004) where our 2010-11 survey data intersects their profile (Ashmore et al., 2020), indicates that the transition between the upper undisrupted high reflectivity ice and the lower deep-ice unit (i.e. R1) can be dated to approximately 16 ka. There are some potential uncertainties associated with this correlation (e.g. resolution of the Byrd ice core, the resolution, geolocation and vertical and along track sampling of the SPRI-NSF-TUD RES data etc.). However,

assuming this correlation to be correct, the low-reflectivity zone between R1 and R2 is of last glacial age, with the potential for even older ice in places."

3. We have removed Figure 1c, which mapped the extent of unit R2. Figure 1d is now 1c.

Dr Neil Ross Newcastle University 4th May 2020

[revised manuscript text omitted]